# MuJoCo Manipulus:
# A Robot Learning Benchmark
# for Generalizable Tool Manipulation

## Abstract

We propose MuJoCo Manipulus, a novel open-source benchmark powered by the MuJoCo physics simulation engine, designed to accelerate advances in robot learning for tool manipulation. Our benchmark includes a diverse set of tasks for tool manipulation —a domain where the field currently lacks a unified benchmark. Different research groups rely on custom-designed tasks or closed-source setups, limiting cross-comparability and hindering significant progress in this field. To that end, our benchmark provides 16 challenging tool manipulation tasks, including variants of Pouring, Scooping, Scraping, Stacking, Gathering, Hammering, Mini-Golf, and Ping-Pong. The benchmark supports both state-based and vision-based observation spaces, is fully integrated with the Gymnasium API, and seamlessly connects with widely used Deep Reinforcement Learning libraries, ensuring easy adoption by the community. We conduct extensive reinforcement learning experiments on our benchmark, and our results demonstrate that there is substantial progress to be made for training tool manipulation policies. Our codebase and additional videos of the learned policies can be found on our anonymous project website: `mujoco-manipulus.github.io`.

## 1 Introduction

Robot learning has recently experienced a rapid transformation, driven by developments in both hardware and algorithms. A fundamental problem in robotics is tool manipulation, where a robot uses an external device to assist itself in accomplishing a manipulation objective. Common tasks (and their tools) include assistive feeding using forks and other utensils (Sundaresan et al., 2023; Jenamani et al., 2024), cutting items (Heiden et al., 2021; Xu et al., 2023b), hammering using hammers (Fang et al., 2018), and scooping using spoons and ladles (Seita et al., 2022; Grannen et al., 2022; Qi et al., 2024). By not limiting a robot to its native gripper hardware, tool manipulation can greatly extend the tasks that robots can perform. More broadly, understanding how to effectively use external tools is often considered a sign of greater intelligence (Baber, 2003; Washburn, 1960). To operate a tool, the robot must reason about the function of the tool, its limitations, and its potential effects on surrounding objects. Furthermore, tools are highly diverse and vary along many axes, including (but not limited to) size, shape, and deformability. Therefore, tool manipulation presents an elusive set of open problems despite tremendous progress in robot learning.

While there has been considerable progress in robot tool manipulation, a core challenge in the field boils down to the lack of a unified tool manipulation benchmark—existing works conduct experiments using different setups and tasks, making it harder to compare algorithms and to measure progress in the field. To our knowledge, such a benchmark does not exist for fair comparison of methods for tool manipulation. While existing manipulation benchmarks such as ManiSkill2 (Gu et al., 2023) and Robosuite (Zhu et al., 2020) contain tasks that involve some tool usage (such as using a scooper to scoop granular media), they are not specialized to tool manipulation and not ideal testbeds for studying the generalization to different tools. In closely-related work,( Holladay et al. (2019)) provides printable tool models and experimental data, supporting robot grasping with certain tools. However, it focuses on open-loop manipulation with parallel-jaw grippers, making it less effective to reflect algorithm performance in the real world. In contrast, this work provides

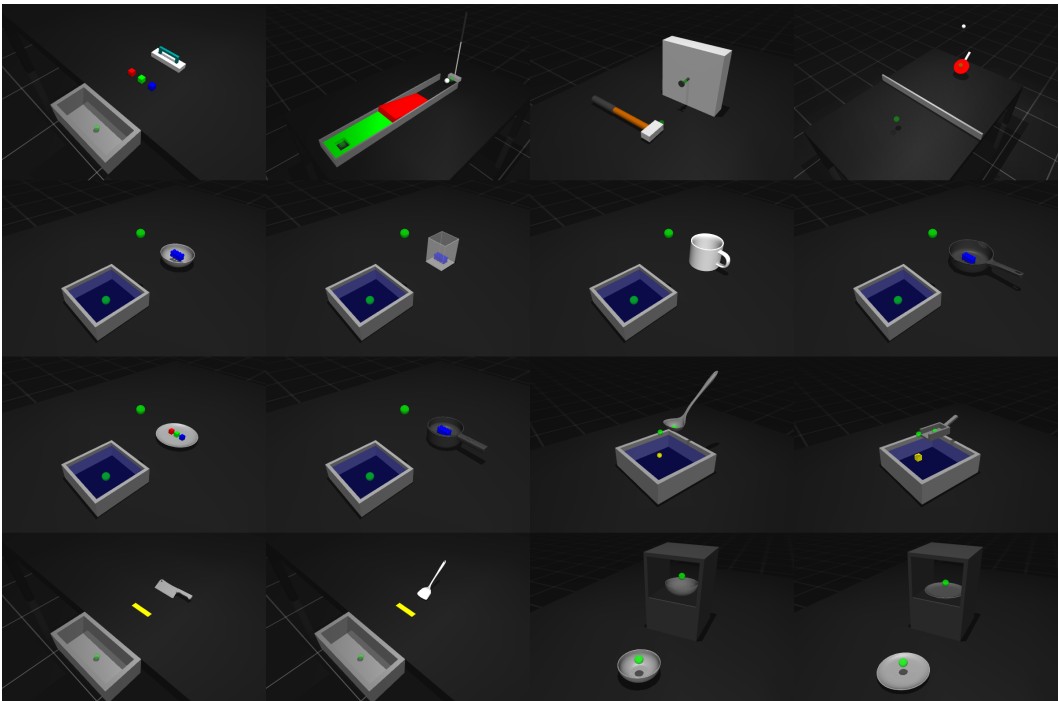

Figure 1: MuJoCo Manipulus includes a diverse set of 16 tool manipulation tasks. We have 8 task categories including `Gathering`, `Mini-Golf`, `Hammering`, `Ping-Pong`, `Pouring`, `Scooping`, `Scraping`, and `Stacking`. Each task provides a unique tool to the user, with a total of 14 tools in our Benchmark, and is integrated with the Gymnasium API (Towers et al., 2024) for benchmarking reinforcement learning algorithms.

a scalable simulation benchmark that focuses on tool manipulation under different scenarios. Our goal is to enable closed-loop policy learning for tool manipulation via reinforcement learning.

Towards this goal, we propose the MuJoCo Manipulus benchmark, built with the MuJoCo physics simulation engine (Todorov et al., 2012). Our benchmark provides an elegant and flexible pipeline for designing simulation environments in MuJoCo, and learning control policies in these environments with the Gymnasium API (Towers et al. (2024)). MuJoCo Manipulus is centered on tool manipulation, where the agent controls a free-floating tool. This design allows future research to begin with simpler setups (free-floating tools) before progressing to more complex variations where tool manipulation must integrate with a robot arm. As part of our benchmark, we rigorously evaluate 3 well-established model-free reinforcement learning algorithms. Our findings show that while these algorithms perform reasonably well on some tasks, they face challenges on certain classes of tasks. These limitations highlight opportunities for future research in robot tool manipulation.

In summary, the contributions of our paper include:

- A novel open-source tool manipulation benchmark, MuJoCo Manipulus, powered by the MuJoCo physics simulation engine, with the following key features:.
  - We provide 16 tool manipulation tasks to the community, with 14 tools, to accelerate research advances in tool manipulation.
  - Our benchmark supports state, RGB, and state+RGB observation spaces, allowing for benchmarking of various reinforcement learning and representation learning methods.
  - Elegant and accessible implementations of MuJoCo-Gymnasium tasks, which can be easily extended to more complex settings, and empower the research community to build additional tasks with our framework.
- Experimental results of 3 well-established model-free reinforcement learning algorithms on our benchmark, showcasing their promise but also limitations, thus motivating questions for future work.

## 2 RELATED WORK

Table 1: **Comparison of Simulation Benchmarks**: We compare our Simulated Tool Manipulation Benchmark to several popular Simulation Benchmarks. For a complete list of Tool Skills for each benchmark, please see our Appendix.

| Benchmark | # of Tasks | Dense Rewards | # of Tool Skills | Simulation Engine |
|---|---|---|---|---|
| Meta-World (Yu et al. (2019)) | 50 | ✓ | 5 | MuJoCo |
| RoboSuite (Zhu et al. (2020)) | 9 | ✓ | 3 | MuJoCo |
| Fleet-Tools (Hoque et al. (2022)) | 4 | ✗ | 4 | Drake |
| ManiSkill2 (Gu et al. (2023)) | 19 | ✓ | 6 | SAPIEN |
| RLBench (James et al. (2020)) | 100 | ✗ | 7 | CoppeliaSim |
| **MuJoCo Manipulus (Ours)** | 16 | ✓ | 8 | MuJoCo |

### 2.1 ROBOT TOOL MANIPULATION

Robot tool manipulation is a decades-old research area (Asada & Asari, 1988) which has seen a recent explosion of interest. Research in the area can be broadly characterized as works that focus on general methods for tool manipulation, versus those that study a specific type of tool manipulation. Among the former category are works that have explored methods for manipulating tools, such as by learning from keypoint (Qin et al., 2020; Turpin et al., 2021) or flow (Seita et al., 2022) representations, using differentiable trajectory optimization (Lin et al., 2022; Qi et al., 2022) or learning dynamics models either through vision (Xie et al., 2019) or contact (Van der Merwe et al., 2022). Researchers have also explored learning to design tools (Liu et al., 2023; Dong et al., 2024) and their morphology (Li et al., 2023). Recently, there has been work on robots that learn to manipulate diverse tools using techniques such as task and motion planning techniques (Wang et al., 2019), trajectory generation (Qi et al., 2024), or large language models (Xu et al., 2023a; Ren et al., 2022).

The second category of works specialize to specific types of tool manipulation tasks, such as scooping (Schenck et al., 2017; Grannen et al., 2022), pouring (Narasimhan et al., 2020; Schenck & Fox, 2017), cutting (Heiden et al., 2021; Xu et al., 2023b), and tools for cooking (Shi et al., 2023). In contrast to these works, which either propose largely tool- or task-specific methods, or which craft a few tasks to test (due to lack of a pre-existing benchmark), our focus is on developing a simulation benchmark for tool manipulation that tests a variety of tasks. We focus on rigid object tool manipulation, since there are a wide number of these tasks that can be designed with MuJoCo and solved with RL. Closer to our paper includes prior work such as (Rajeswaran et al., 2018), which uses free-floating dexterous hands to manipulate the objects, but instead focuses on object reorientation instead of tool manipulation. In addition, (Wang et al., 2024) open-source four tool manipulation tasks powered by Drake simulation (Tedrake & the Drake Development Team, 2019), but their focus is on developing algorithms for learning from fleets of robots instead of benchmark development to support future research in tool manipulation and reinforcement learning. In contrast, we present a wider-scale tool manipulation benchmark with substantially more tasks, tool skills, and dense rewards for all task categories.

### 2.2 BENCHMARKS IN ROBOT LEARNING AND MANIPULATION

Benchmarks have played a critical role in the advancement of robot learning research by facilitating comparisons among policy-learning methods, and providing insights for future research areas. Benchmarks can include algorithm implementations or a set of tasks (or both). Examples of high-quality reinforcement learning algorithm implementations include CleanRL (Huang et al., 2022) and Stable Baselines3 (Raffin et al., 2021). Our benchmark is complementary to these algorithms, since we can use reinforcement learning methods to potentially solve each of our tasks.

For manipulation, the community has developed a number of simulation benchmarks. Prominent examples for general manipulation include ManiSkill2 (Gu et al., 2023), RoboSuite (Zhu et al., 2020), and RLBench (James et al., 2020). Other benchmarks focus on meta-learning (Yu et al., 2019) or language-conditioned learning (Mees et al., 2022). Researchers have also created benchmarks

Figure 2: MuJoCo Manipulus provides a general-purpose framework for building Deep Reinforcement Learning tasks on top of the MuJoCo Physics Engine. Above, we demonstrate the simplicity of our pipeline, enabling users to collect diverse tool meshes, design their tasks, train deep reinforcement learning policies (e.g., using `CleanRL` (Huang et al., 2022)), and evaluate them.

specializing in domains as diverse as tabletop rearrangement (Zeng et al., 2020), deformable object manipulation (Lin et al., 2020; Seita et al., 2021), fetching (Han et al., 2024), fleet learning (Hoque et al., 2022), navigation and manipulation in homes (Nasiriany et al., 2024; Szot et al., 2021; Li et al., 2022), and surgical robotics (Richter et al., 2019; Schmidgall et al., 2024; Yu et al., 2024). Recent benchmarks for higher-DOF manipulation include those focusing on piano playing (Zakka et al., 2023) and training humanoids (Sferrazza et al., 2024). BiGym (Chernyadev et al., 2024) and SMPLOlympics (Luo et al., 2024b) also provide simulated humanoid tasks, some of which have tool manipulation (e.g., flipping a sandwich with a spatula). Finally, other benchmarks study complementary areas such as real-world furniture assembly (Heo et al., 2023) and generalizable manipulation (Luo et al., 2024a). While these benchmarks have been crucial to the robot learning community, none specialize in tool manipulation, and not all of them provide dense rewards for reinforcement learning. Our MuJoCo Manipulus benchmark thus fills a critical need in the robot learning community. In addition, our work is up-to-date with the latest software advances introduced in MuJoCo 3.0+ (Todorov et al., 2012), and follows the newer Gymnasium interface (Towers et al., 2024) instead of the OpenAI gym interface (Brockman et al., 2016).

## 3 THE BENCHMARK: MUJOCO MANIPULUS

We formally introduce our benchmark, MuJoCo Manipulus. In Section 3.1, we first outline some general principles we follow for the benchmark, plus different features we support. Then, we discuss our tool manipulation tasks in Section 3.2.

### 3.1 OVERVIEW OF SIMULATION FRAMEWORK

We support several important features in MuJoCo Manipulus which makes it a desirable long-term benchmark for the robot learning community. Our benchmark provides users with a flexible pipeline for building end-to-end reinforcement learning tasks (i.e., environments) on top of the MuJoCo physics simulation engine. We decouple our pipeline into four steps which streamline the development process for RL environments (see Figure 2).

**Collecting Tool Meshes**: The first step is to collect tool meshes from diverse data sources. Our tools are either hand-designed in MuJoCo with built-in shape primitives, or are hand-selected from

the TACO Dataset (Liu et al. (2024)) or MuJoCo's open-source models (Todorov et al. (2012)). In total, our benchmark contains 14 tools, with in-category object variants for 4 of our task categories.

**Designing our Tasks**: In the first stage of task design, we generate collision meshes for tools from the TACO Dataset by performing a convex decomposition of each object with CoACD (Wei et al. (2022)). For tools that are hand-designed in MuJoCo with built-in shape primitives, or are from MuJoCo's open-source models, this step is not necessary.

The second stage involves creating a task XML file with MuJoCo that contains task-relevant objects. We define each object, including our tools, in object XML files. Subsequently, we perform relative imports of the object XML files into our task XML files to construct the simulation scene for a given task.

In the third stage, we import our task XML files into Python with PyMJCF, and create Gymnasium (Towers et al. (2024)) environments to encapsulate each task as a Markov Decision Process (MDP). The Gymnasium API provides the framework for the MDP, while MuJoCo provides a Python API for interacting with simulation models and their data. Prior works (Zhu et al. (2020) Yu et al. (2019)) have introduced modular APIs that unify the two interfaces of Gymnasium and MuJoCo. Our work introduces elegant single-file implementations of MuJoCo-Gymnasium tasks, a highly beneficial feature for RL practitioners who can benefit from having access to open-source end-to-end simulation tasks.

Tied into the third stage, our fourth stage involves careful design of task observation spaces, action spaces, environment resets, and dense rewards. A key distinction for tool manipulation tasks is the need for constrained action spaces. Prior works (Seita et al. (2022); Xu et al. (2023b)) have applied constraints to tools so they are compliant, safe, and able to complete their given tasks. In our work, each of our tasks has a constrained action space that is less than 6 DoF, despite MuJoCo supporting 6 DoF control of free-floating objects. Our environment observation spaces include object positions and velocities, as well as tool marker positions and goal positions. Similar to prior simulation benchmark (Gu et al. (2023), Yu et al. (2019), "markers" are free-floating colored spheres in the environment that denote keypoints for a given task. These keypoints are especially useful for dense reward design, and with the use of tool marker positions and goal positions, we were able to write reward functions that generalized across in-category tool variants for each task. For environment resets, we randomize positions of the tools and their goals.

**Training and Evaluating RL Policies**: Our benchmark supports state, RGB, and state+RGB observations for training RL policies. While state-based observations are generally only practical in simulation, they can be useful for simulation-to-real transfer. Such example use cases include asymmetric actor-critic algorithms where the critic is trained with state information (Pinto et al., 2018), and teacher-student distillation algorithms where a "teacher" trains in simulation on state information and a "student" learns to imitate the teacher using only vision-based information (e.g., (Chen et al., 2019; Yuan et al., 2024)).

We apply three well-established model-free reinforcement learning algorithms in our benchmark: **CrossQ** (Bhatt et al. (2024)), Soft Actor-Critic (**SAC**) (Haarnoja et al. (2018)), and Proximal Policy Optimization (**PPO**) (Schulman et al. (2017)). Our SAC and PPO implementations are directly sourced from CleanRL (Huang et al. (2022)), a library of reliable single-file RL algorithm implementations. Our CrossQ implementation is re-implemented from Stable-Baselines3 (Raffin et al. (2021)) in the style of CleanRL, and we found that the performance of CrossQ was consistent with the metrics reported in their paper. Policy training results are logged to Weights and Biases (Biewald (2020)), a popular MLOps tool for logging machine learning experiment results. When a policy is finished training on any of our tasks, we save the final model to Weights and Biases, which can be downloaded from their servers and evaluated locally.

## 3.2 TASKS

For our initial release of MuJoCo Manipulus, we provide 16 tool manipulation tasks and 14 tools. The tools we provide can be broken down into 3 general categories: Kitchen Tools, Home Tools, and Sports Tools. Our "Kitchen Tools" category includes models for Bowl, Mug, Knife, Pan, Pot, Plate, Spatula, Ladle, and Cup objects. Our "Home Tools" category includes models for a Brush, Hammer, and Scooper. Our "Sports Tools" category includes models for a Ping-Pong Paddle and Golf Club.

The tools are either hand-designed by us, taken directly from MuJoCo's open-source object models, or hand-picked from the recent open-source TACO dataset (Liu et al., 2024), which provides tool object meshes and tool interaction data to facilitate understanding bimanual human-tool interactions from video.

In the following, we discuss the tasks in MuJoCo Manipulus. See Figure 1 for visualizations of our 16 tasks. The tasks involve different action spaces, some of which involve rotations. All tool rotations are centered at the "centroid" of the tool's 3D structure, which is equivalently its `MoCap` body location.

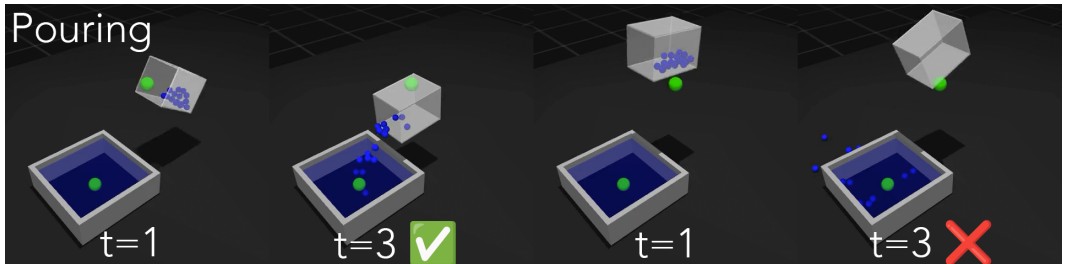

Figure 3: Success and Failure Modes for `Pouring`.

**Pouring Tasks.** In these tasks, the agent controls a tool that starts with 16 particles in it. The agent must use this tool and pour the particles so that they land inside a bin. The action space is 4D, where we allow for changes in the $(x, y, z)$ position and about one axis $\theta_y$ of the tool. The state observation has dimension $\mathcal{S} \in \mathbb{R}^{11}$, with the values consisting of the 3D position of the tool, 1D orientation of the tool, 3D translational velocity of the tool, 1D rotational velocity of the tool, and 3D position of the lifting target marker. A success is when all 16 of the particles land inside of the bin. We support the following variants of pouring:

(1) `PourCup`, using a hand-designed Cup with 16 particles to be poured.
(2) `PourMug`, using the Mug model from MuJoCo's open-source object models, with 16 particles to be poured.
(3) `PourPan`, using the Pan model from TACO, with 16 particles to be poured.
(4) `PourPot`, using the Pot model from TACO, with 16 particles to be poured.
(5) `PourBowl`, using a Bowl model from TACO, with 16 particles to be poured.
(6) `PourPlate`, using a Plate model from TACO. This task has 3 cube particles instead of 16 spherical particles since the plate is flatter compared to the other tools.

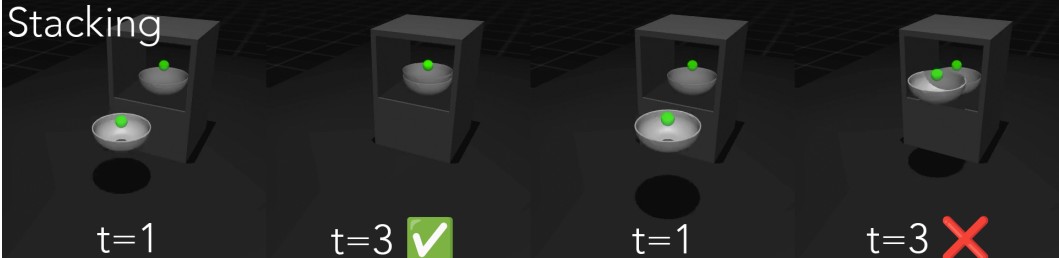

Figure 4: Success and Failure Modes for `Stacking`.

**Stacking Tasks.** In these tasks, the agent must learn to place either a bowl or a plate on top of a static bowl or plate, respectively. The agent's action space is 3D where we allow for changes in the $(x, y, z)$ position of the tool. The state observation has dimension $\mathcal{S} \in \mathbb{R}^{15}$, with the values consisting of the 3D position of the tool, 3D position of the tool marker, 2D velocity of the tool, 1D rotation of the tool, 3D position of the target object, and 3D position of the target object marker. A success is when the plate or bowl which our agent controls has overlap between its "marker" and the

static bowl's "target object marker." Incidentally, we consider bowls and plates as tools since they enable carrying of food and other items, similar to tools like those used in our Pouring tasks.

(7) `StackBowls`, using a Bowl model from TACO.
(8) `StackPlates`, using a Plate model from TACO.

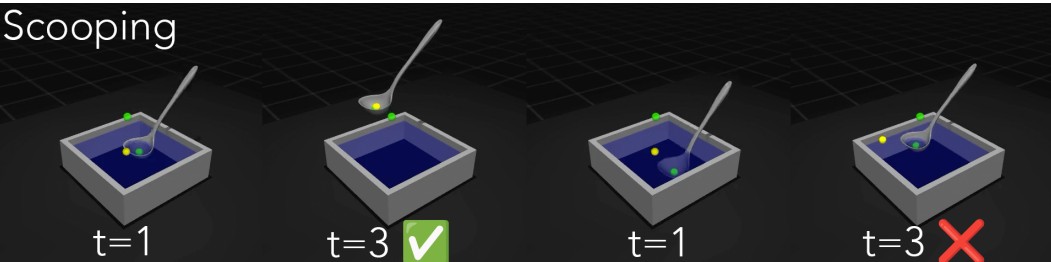

Figure 5: Success and Failure Modes for `Scooping`.

**Scooping Tasks.** In these tasks, the agent controls a scooper-style tool and needs to scoop up a single spherical particle or cube particle from a bin receptacle. The agent's action space is 3D where we allow for changes in the $(x, y, z)$ position of the tool. The state observation has dimension $\mathcal{S} \in \mathbb{R}^{15}$, with the values consisting of the 3D position of the tool, 3D velocity of the tool, 3D position of the tool marker, 3D position of the particle, and 3D position of the lift target marker. A success is when the scooper-style tool has both scooped up the particle and lifted it towards a "target object marker" above the bin receptacle. Our Scooping tasks can be considered as "inverse" versions of our Pouring tasks, and for this reason these tools share similar action spaces.

(9) `ScoopParticles`, using a Ladle model from TACO.
(10) `ScoopCubes`, using a scooper hand-designed with MuJoCo's built-in shape primitives. The tool is similar to the one used in the scooping task from Liu et al. (2023).

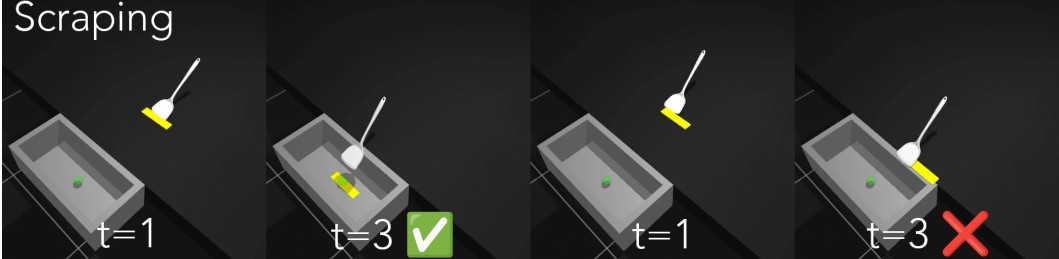

Figure 6: Success and Failure Modes for `Scraping`.

**Scraping Tasks.** In these tasks, we use a kitchen tool to scrape a thin rigid object into a bin receptacle. The agent's action space is 2D where we allow for changes in the $(x, y)$ position of the tool. The state observation has dimension $\mathcal{S} \in \mathbb{R}^{17}$, with the values consisting of the 3D position of the tool, 2D velocity of the tool, 3D position of the MoCap object moving the tool, 3D position of the tool marker, 3D position of the thin rigid object, and 3D position of the bin target marker. A success is when the thin rigid object is scraped and lands inside the bin receptacle.

(11) `ScrapeKnife`, using a Knife model from TACO.
(12) `ScrapeSpatula`, using a Spatula model from TACO.

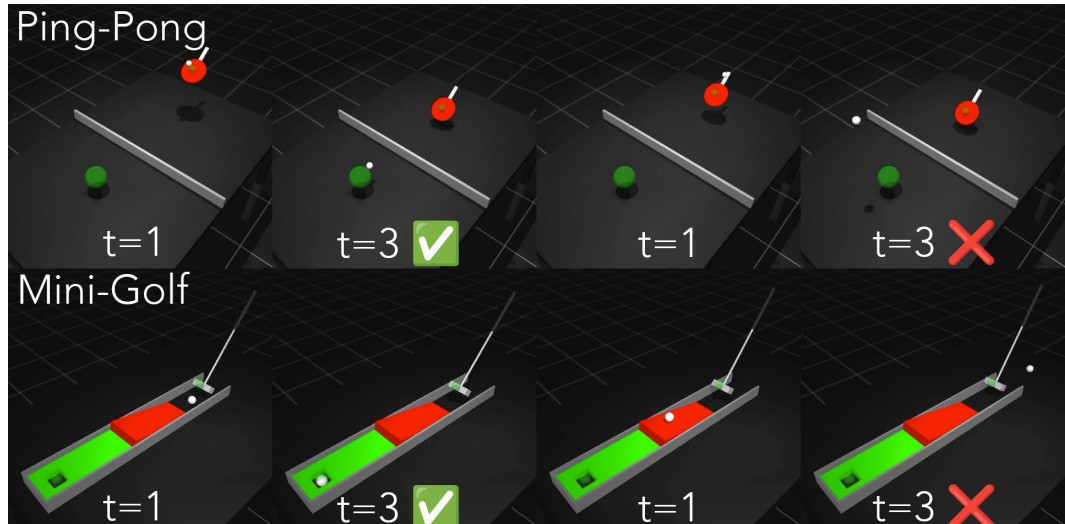

Figure 7: Success and Failure Modes for `Ping-Pong` and `Mini-Golf`.

**Sports Tasks.** We include two sports tasks that require tool usage in our benchmark.

(13) `Ping-Pong` uses a hand-designed ping-pong paddle with a handle as the tool. The objective is for the agent to to track a ping-pong ball in mid-air and hit it towards the opposite end of the table. The agent's action space is 2D where we allow for changes in the $(x, z)$ position of the tool. The state observation has dimension $\mathcal{S} \in \mathbb{R}^{21}$, with the values consisting of the 3D position of the paddle tool marker, 6D velocity of the paddle, 3D position of the ball, 6D velocity of the ball, and 3D position of the target marker. A success is when the ball hits the opposite end of the table within $\epsilon < 0.1$ distance of the marker.

(14) `Mini-Golf` uses a hand-designed golf club as the tool. The objective is for the agent to hit a golf ball into a hole. The agent's action space is 2D where we allow for changes in the $(x, )$ position and about one axis $\theta_y$ of the tool. The state observation has dimension $\mathcal{S} \in \mathbb{R}^{28}$, with the values consisting of the golf club's 3D position, 4D orientation, 6D velocity, the golf club's green marker, the ball's 3D position and 6D velocity, and the target hole's 3D position. A success is when the ball lands inside the hole.

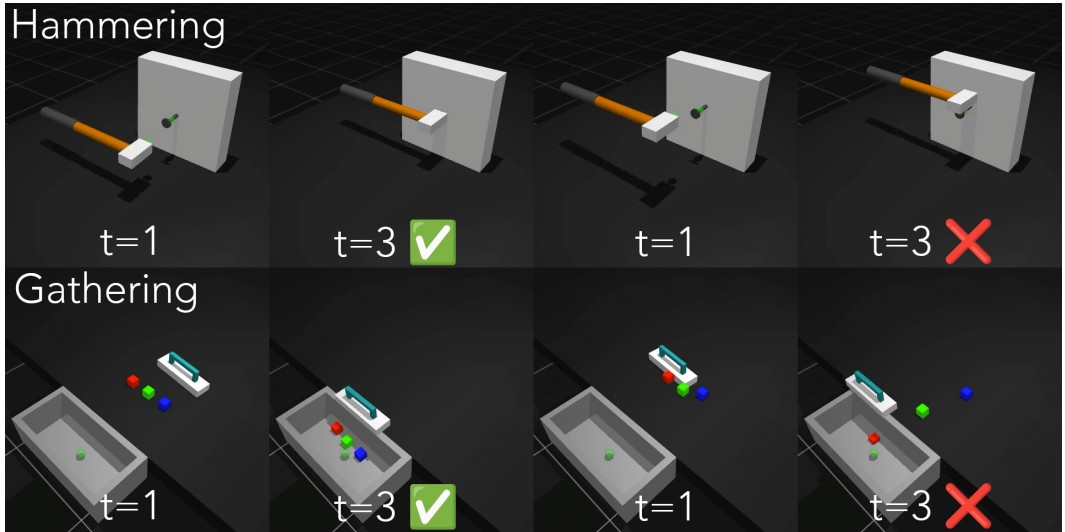

Figure 8: Success and Failure Modes for `Hammering` and `Gathering`.

**Miscellaneous Tasks.** These remaining tool manipulation tasks do not fall under a clear category.

(15) `HammerNail` uses a hand-designed hammer as the tool. The objective is for the agent to push a nail into a box. The agent's action space is 2D where we allow for changes in the $(x, z)$ position of the tool. The state observation has dimension $\mathcal{S} \in \mathbb{R}^{14}$, with the values consisting of the 3D position of the hammer, 2D velocity of the hammer, 3D position of the hammer marker, 3D position of the initial nail target marker, and 3D position of the final nail target marker. The nail has 1 degree of freedom to enable forward and backward movement into the box. A success is when the nail has been fully pushed into the box.

(16) `GatherCube` uses a hand-designed brush with a handle as the tool. The objective is for the agent to gather three multi-colored cubes and push them inside of a receptacle bin with an open front. The agent's action space is 2D where we allow for changes in the $(x, y)$ position of the tool. The state observation has dimension $\mathcal{S} \in \mathbb{R}^{17}$, with the values consisting of the 3D position of the brush, 2D velocity of the brush, 3D positions of each of the 3 cube particles, and the 3D position of the target marker inside the bin. A success is when all three cubes are gathered and inside the bin receptacle.

## 4 LEARNING ROBOT TOOL MANIPULATION

### 4.1 REINFORCEMENT LEARNING EXPERIMENTS

To learn the proposed tool manipulation tasks in MuJoCo Manipulus, we train reinforcement learning policies using **CrossQ** (Bhatt et al. (2024)), **SAC** (Haarnoja et al. (2018)), and **PPO** (Schulman et al. (2017)), which are 3 well-established model-free reinforcement learning algorithms used by the robot learning community. We measure the performance of each method with state inputs, resulting in three distinct results that show the upper-bound performance of model-free RL methods on our benchmark. We train task-specific policies, and leave multi-task learning to future work. Each reinforcement learning run lasts for either 100,000 or 300,000 training steps. In the case of `Stacking` and `Scooping` tasks, we train for 300,000 steps since we empirically found that these tasks are more difficult to learn than our other tasks. While we provide users with sparse and dense rewards, we benchmark using the dense reward functions to enable better guidance for reinforcement learning baselines. We run 5 seeds per task, and provide the averaged success rate curves with 95% CI shading for each task. See Figure 9 for success rate results on our tasks.

### 4.2 REINFORCEMENT LEARNING RESULTS

Overall, we find reasonable success rates for all our tasks. The easiest tasks to learn in our benchmark are `HammerNail`, `Pour Plate`, `Scrape Spatula`, `Scrape Knife`, and `Gather Cubes`, which have simpler action spaces and objects for the tools to interact with. The best-performing baseline method is CrossQ, which is a more sample-efficient version of Soft Actor-Critic. In general, CrossQ and SAC were our best-performing baselines because they are off-policy methods, and off-policy methods are known to be more sample-efficient than on-policy methods like PPO. However, there were unique instances where SAC and PPO performed better than CrossQ. SAC performed best in the `Mini Golf` task, and PPO performed best in the `Ping Pong` task. A possible reason for this is because we found CrossQ overfits to high-reward states it encounters early in training, whereas PPO and SAC do not overfit to early training experiences, even if they yield high rewards.

Our benchmark's harder tasks include `Scoop Particle`, `Scoop Cube`, `Stack Plates`, `Stack Bowls`, and `Ping Pong`. For most of our tasks, we used a frame skip value of 12, but had to reduce our frame skip value to 5 for Ping Pong to allow the paddle enough time to reach the ball and hit it. Stacking tasks are subject to the plate and bowl not perfectly aligning with the static plate or bowl, which is considered a failure case. Scooping tasks are difficult in 100K training steps, but we found that additional training time allows CrossQ to learn good policies in both settings. We provide qualitative results for all tasks on our project site, and include visualizations of success and failures for each task category in the prior section. Overall, we found that reinforcement learning methods with constrained action spaces provide a promising interface for learning tool manipulation.

Our benchmark is also reasonably fast, with 100K training steps taking 10 minutes of walltime, and 300K training steps taking 30 minutes of walltime. These measurements are reported with

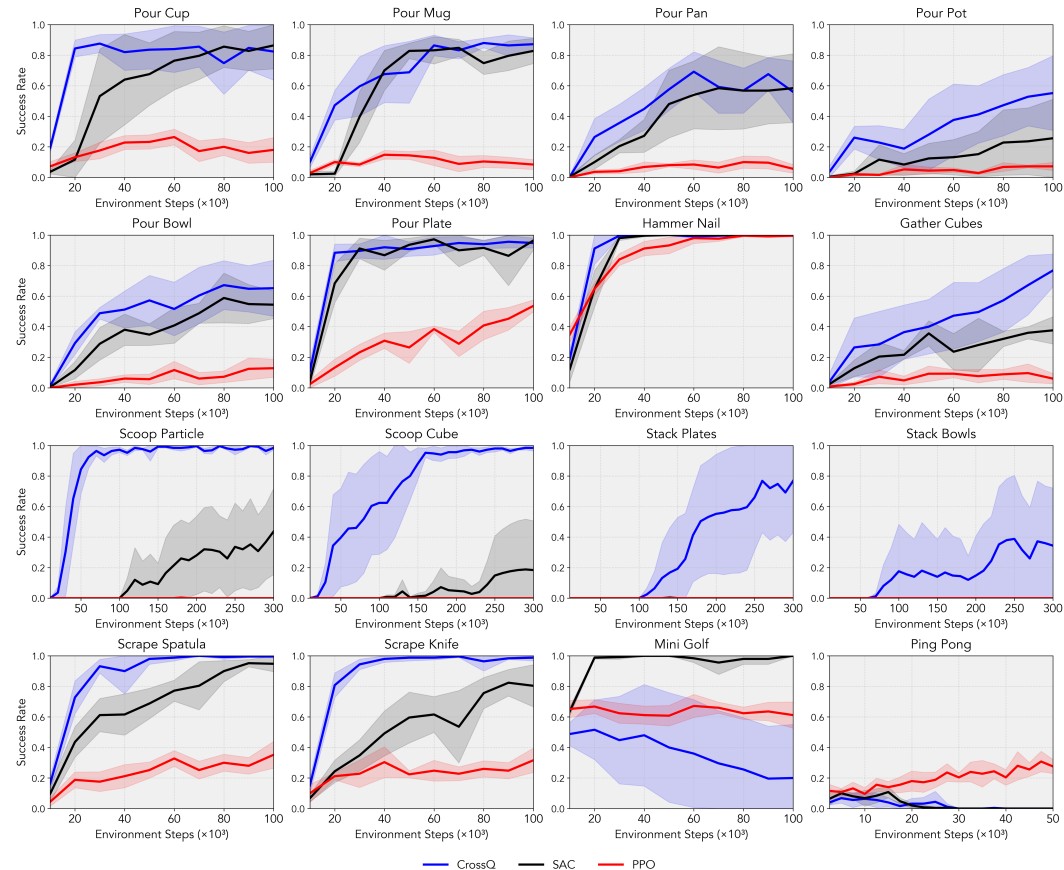

Figure 9: Success rate curves for the 16 tasks in MuJoCo Manipulus with 3 Model-Free RL Baselines: CrossQ, SAC, and PPO. Success Rates are reported every 50 episodes as the average success over the last 50 episodes, and each baseline result is averaged over 5 seeds with shading for the 95% CIs. We do not apply smoothing to the curves.

an NVIDIA RTX 4090 GPU and an Intel i9-13900K CPU. In the future, we plan to integrate the recent MuJoCo-XLA (MJX) bindings into our benchmark so our simulation can run even faster with GPU-accelerated physics simulation.

## 5 CONCLUSION

This paper proposes a novel benchmark, MuJoCo Manipulus, which contains 16 tool manipulation tasks that collectively include 14 diverse tools. We benchmark CrossQ, SAC, and PPO for learning tool manipulation. Our findings reveal that there are multiple tasks where these methods struggle to learn successful policy behaviors. This motivates directions in future work to improve robot tool manipulation.

While promising, MuJoCo Manipulus has a few limitations. Our benchmark runs simulation on the CPU, and in the future we plan to integrate MuJoCo-XLA (MJX) support for accelerated physics simulation on the GPU. Additional improvements to our work include: supporting data collection and imitation learning; evaluating simulation-to-real transfer capability of algorithms developed with our benchmark; and incorporating bimanual and dexterous robot manipulation. We hope this inspires a new era in robot learning and tool manipulation.

**Ethics Statement.** This paper does not involve the collection or annotation of new data. We build this benchmark on top of a well-established simulator—MuJoCo—which is released under strict

ethical guidelines. While we do not see any immediate ethics concerns, we acknowledge that our research could be used as part of an eventual robot system that abuses tool use. For autonomous robots, it is critical to ensure their safety when they perform delicate or dangerous manipulation tasks, especially in the presence of humans. We strive to ensure that our benchmark, as well as future applications on top of this benchmark, are developed responsibly and ethically to maintain safety and preserve privacy.

**Reproducibility Statement.** MuJoCo Manipulus is built based on the well-established MuJoCo physics engine with enhanced user support. We will fully release our code and scripts to accurately reproduce the results in this paper. We commit to providing first-class support for future robot learning research. In addition, we plan to continue improving the benchmark by expanding the range of tool manipulation tasks and other robot learning tasks.

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

# A    ADDITIONAL DETAILS OF SIMULATION TASKS

## A.1    COMPARISON OF TOOL SKILLS IN CURRENT BENCHMARKS

We compare the distribution of Tool Skills in MuJoCo Manipulus with other benchmarks, and include the # of Tool Skills in other benchmarks below:

**Meta-World**

| Skills | Tools |
|---|---|
| Assembly | Ring Tool |
| Disassembly | Ring Tool |
| Hammering | Hammer |
| Insertion | Peg |
| Removal | Peg |

**RoboSuite**

| Skills | Tools |
|---|---|
| Assembly | Peg/Nut |
| Wiping | Brush |
| Insertion | Peg |

**Fleet-Tools**

| Skills | Tools |
|---|---|
| Scooping | Spatula |
| Splitting | Knife |
| Hitting | Hammer |
| Spanning | Wrench |

**ManiSkill2**

| Skills | Tools |
|---|---|
| Insertion | Peg |
| Plugging | Charger |
| Filling | Bucket |
| Excavating | Shovel |
| Pouring | Bottle |
| Writing | Pencil |

**RLBench**

| Skills | Tools |
|---|---|
| Closing | Box, Door, Drawer, Fridge, Grill, Jar, Laptop, Microwave |
| Emptying | Container, Dishwasher |
| Hitting | Pool Cue, Hockey |
| Insertion | Peg, USB, Charger |
| Pick-and-Place | Cup, Plate |
| Sweeping | Broom, Dustpan |
| Removal | USB |

**MuJoCo Manipulus (Ours)**

| Skills | Tools |
|---|---|
| Pouring | Cup, Mug, Pan, Pot, Bowl, Plate |
| Stacking | Bowl, Plate |
| Scooping | Ladle, Hand-Shovel |
| Scraping | Spatula, Butcher Knife |
| Ping-Pong | Paddle |
| Mini-Golf | Golf Club |
| Hammering | Hammer |
| Gathering | Brush |

## A.2 OBSERVATION SPACE

In our tasks, we support state and visual observations. During experiments, we use state-based observations to train each baseline method. The size of visual observations is $3 \times 128 \times 128$, representing a 128x128 RGB image of the environment. The observation spaces for each task are described in the main text.

## A.3 ACTION SPACE

In our tasks, we provide 2-DoF, 3-DoF, or 4-DoF action spaces by default, which are carefully chosen to ensure the tool is capable of solving the given task while exhibiting safe, compliant tool behavior. However, all our tasks can, in principle, be extended to full 6-DoF action spaces by enabling the full degrees of rotation. We restrict the number of DoFs mainly to enable off-the-shelf RL algorithms to make reasonable learning progress. The action spaces for each task are described in the main text.

## A.4 REWARD FUNCTIONS

Our benchmark supports both sparse and dense rewards. In the main paper, we benchmark using dense rewards since it is necessary to guide standard reinforcement learning algorithms. However, we encourage the community to explore learning from sparse rewards. Below, we provide details of our reward functions.

### A.4.1 PRELIMINARIES

We use a tolerance function, originally from the DeepMind Control Suite (Tunyasuvunakool et al. (2020)), to constrain individual rewards to the range $[0, 1]$ while applying smooth increases and decreases to those rewards w.r.t. changes in the environment. We empirically found that using tolerance functions for individual rewards leads to more stable policy learning. Related benchmarks, such as Meta-World (Yu et al. (2019)) and ManiSkill2 (Gu et al. (2023)), also use a similar notion of tolerance functions for their rewards.

**Pouring Tasks.**

The reward function consists of three stages:

STAGE 1: REACH THE LIFT TARGET

Let the tool position be $\mathbf{p}_{\text{tool}}$ and the lift target position be $\mathbf{p}_{\text{lift\_target}}$. Define the distance between these positions as:

$$d = \|\mathbf{p}_{\text{tool}} - \mathbf{p}_{\text{lift\_target}}\|$$

The lift reward is given by:

$$R_{\text{lift}} = \text{tolerance}(d, \text{bounds} = [0, 0.05], \text{margin} = 0.1, \text{sigmoid} = \text{gaussian})$$

STAGE 2: ROTATE THE TOOL

Let the tool's rotation about the relevant axis be $q_{\text{tool}}$. Define the bounds for a valid rotation as $[0.7, 0.9]$. The pour reward is:

$$R_{\text{pour}} = \text{tolerance}(q_{\text{tool}}, \text{bounds} = [0.7, 0.9], \text{margin} = 0.7, \text{sigmoid} = \text{gaussian})$$

STAGE 3: CHECK PARTICLES IN BIN

Let the bin target position be $\mathbf{p}_{\text{bin\_target}}$, and the positions of particles be $\mathbf{p}_i$ for $i = 1, \ldots, N$. Compute the distances from particles to the bin target:

$$d_i = \|\mathbf{p}_i - \mathbf{p}_{\text{bin\_target}}\|$$

If $R_{\text{lift}} = 1.0$, the bin reward is:

$$R_{\text{bin}} = \frac{1}{N} \sum_{i=1}^{N} \text{tolerance}(d_i, \text{bounds} = [0, 0.09], \text{margin} = 0, \text{sigmoid} = \text{gaussian})$$

Otherwise:

$$R_{\text{bin}} = 0$$

TOTAL REWARD

The total reward is:

$$R = R_{\text{lift}} + R_{\text{pour}} + R_{\text{bin}}$$

The reward is clipped to the range $[0, 3]$:

$$R = \min(\max(R, 0), 3)$$

SUCCESS CONDITION FOR SPARSE REWARD

The task is considered successful if:

$$\text{success} = \begin{cases} 1 & \text{if } R_{\text{bin}} = 1.0 \\ 0 & \text{otherwise} \end{cases}$$

**Stacking Tasks.**

The reward function is composed of the following stages:

STAGE 1: MINIMIZE DISTANCE BETWEEN MARKER AND TARGET

Let the marker position be $\mathbf{p}_{\text{marker}}$ and the target position be $\mathbf{p}_{\text{target}}$. Define the distance between these positions as:

$$d_{\text{a}} = \|\mathbf{p}_{\text{marker}} - \mathbf{p}_{\text{target}}\|$$

The reward for minimizing this distance is:

$$R_{\text{stage\_1}} = \text{tolerance}(d_{\text{a}}, \text{bounds} = [0, 0.01], \text{margin} = 0.275, \text{sigmoid} = \text{gaussian})$$

### STAGE 1 PENALTY: ENSURE QUATERNION STABILITY

Let the first component of the quaternion for the marker body be $q_0$. To ensure stability, we compute:

$$P_{\text{quat}} = \text{tolerance}(q_0, \text{bounds} = [0.99, 1.01], \text{margin} = 0.01, \text{sigmoid} = \text{gaussian})$$

### TOTAL REWARD

The total reward is computed as the product of the stage 1 reward and the quaternion penalty:

$$R = R_{\text{stage\_1}} \cdot P_{\text{quat}}$$

If $R = 1.0$, an additional reward of $1.0$ is added:

$$R = \begin{cases} R + 1.0 & \text{if } R = 1.0 \\ R & \text{otherwise} \end{cases}$$

Finally, the reward is clipped to the range $[0, 2]$:

$$R = \min(\max(R, 0), 2)$$

### SUCCESS CONDITION FOR SPARSE REWARD

The task is considered successful if:

$$\text{success} = \begin{cases} 1 & \text{if } R = 1.0 \\ 0 & \text{otherwise} \end{cases}$$

**Scooping Tasks.**

The reward function consists of two stages:

### STAGE 1: REACH THE YELLOW PARTICLE

Let the position of the scooper be $\mathbf{p}_{\text{scooper}}$, and the position of the yellow particle be $\mathbf{p}_{\text{particle\_yellow}}$. Define the horizontal distance between these positions as:

$$d_{\text{a}} = \|\mathbf{p}_{\text{scooper}} - \mathbf{p}_{\text{particle\_yellow}}\|$$

The reward for reaching the yellow particle is:

$$R_{\text{reach}} = \text{tolerance}(d_{\text{a}}, \text{bounds} = [0, 0.025], \text{margin} = 0.1, \text{sigmoid} = \text{gaussian})$$

Additionally, introduce a penalty based on the height difference between the scooper and the yellow particle:

$$h_{\text{diff}} = p_{\text{scooper},z} - p_{\text{particle\_yellow},z}$$

The height penalty is given by:

$$P_{\text{height\_a}} = \text{tolerance}(h_{\text{diff}}, \text{bounds} = (-\infty, 0], \text{margin} = 0.02, \text{sigmoid} = \text{gaussian})$$

The adjusted reach reward is:

$$R_{\text{reach}} = R_{\text{reach}} \cdot P_{\text{height\_a}}$$

### STAGE 2: SCOOP THE YELLOW PARTICLE TOWARDS THE LIFT TARGET

Let the position of the lift target be $\mathbf{p}_{\text{target}}$. Define the distance between the yellow particle and the lift target as:

$$d_{\text{b}} = \|\mathbf{p}_{\text{particle\_yellow}} - \mathbf{p}_{\text{target}}\|$$

The reward for scooping the yellow particle towards the lift target is:

$$R_{\text{scoop}} = \text{tolerance}(d_{\text{b}}, \text{bounds} = [0, 0.05], \text{margin} = 0.05, \text{sigmoid} = \text{gaussian})$$

Introduce a penalty based on the height difference between the scooper and the lift target:

$$h_{\text{diff\_b}} = p_{\text{scooper},z} - p_{\text{target},z}$$

The height penalty for this stage is:

$$P_{\text{height\_b}} = \text{tolerance}(h_{\text{diff\_b}}, \text{bounds} = [0, 0.1], \text{margin} = 0.1, \text{sigmoid} = \text{gaussian})$$

The adjusted scoop reward is:

$$R_{\text{scoop}} = R_{\text{scoop}} \cdot P_{\text{height\_b}}$$

TOTAL REWARD

The total reward is the sum of the reach and scoop rewards:

$$R = R_{\text{reach}} + R_{\text{scoop}}$$

If $R_{\text{scoop}} \geq 0.95$, a success bonus of 2.0 is added:

$$R = \begin{cases} R + 2.0 & \text{if } R_{\text{scoop}} \geq 0.95 \\ R & \text{otherwise} \end{cases}$$

Finally, the reward is clipped to the range $[0, 4]$:

$$R = \min(\max(R, 0), 4)$$

SUCCESS CONDITION FOR SPARSE REWARD

The task is considered successful if:

$$\text{success} = \begin{cases} 1 & \text{if } R_{\text{scoop}} \geq 0.95 \\ 0 & \text{otherwise} \end{cases}$$

**Scraping Tasks.**

The reward function consists of two stages:

STAGE 1: REACHING REWARD

Let the position of the tool be $\mathbf{p}_{\text{tool}}$, and the position of the yellow particle be $\mathbf{p}_{\text{particle\_yellow}}$. Define the distance between these positions as:

$$d_{\text{reach}} = \|\mathbf{p}_{\text{tool}} - \mathbf{p}_{\text{particle\_yellow}}\|$$

The reaching reward is given by:

$$R_{\text{reach}} = \text{tolerance}(d_{\text{reach}}, \text{bounds} = [0, 0.01], \text{margin} = 0.12, \text{sigmoid} = \text{gaussian})$$

STAGE 2: MOVING TO BIN REWARD

Let the position of the bin target be $\mathbf{p}_{\text{bin\_target}}$. Define the distance between the yellow particle and the bin target as:

$$d_{\text{move}} = \|\mathbf{p}_{\text{particle\_yellow}} - \mathbf{p}_{\text{bin\_target}}\|$$

The moving reward is given by:

$$R_{\text{move}} = \text{tolerance}(d_{\text{move}}, \text{bounds} = [0, 0.05], \text{margin} = 0.1825, \text{sigmoid} = \text{gaussian})$$

TOTAL REWARD

The total reward is the sum of the reaching reward and the moving reward:

$$R = R_{\text{reach}} + R_{\text{move}}$$

If $R_{\text{move}} = 1.0$, indicating that the yellow particle is successfully in the bin, a success bonus of 2.0 is added:

$$R = \begin{cases} R + 2.0 & \text{if } R_{\text{move}} = 1.0 \\ R & \text{otherwise} \end{cases}$$

Finally, the reward is clipped to the range $[0, 4]$:

$$R = \min(\max(R, 0), 4)$$

SUCCESS CONDITION FOR SPARSE REWARD

The task is considered successful if:

$$\text{success} = \begin{cases} 1 & \text{if } R_{\text{move}} = 1.0 \\ 0 & \text{otherwise} \end{cases}$$

**Ping Pong.**

The reward function consists of two stages:

STAGE 1: MINIMIZE DISTANCE BETWEEN BALL AND PADDLE

Let the position of the paddle (marker) be $\mathbf{p}_{\text{paddle}}$ and the position of the ball be $\mathbf{p}_{\text{ball}}$. Define the distance between these positions as:

$$d_{\text{a}} = \|\mathbf{p}_{\text{paddle}} - \mathbf{p}_{\text{ball}}\|$$

The reward for minimizing this distance is:

$$R_{\text{stage\_1}} = \text{tolerance}(d_{\text{a}}, \text{bounds} = [0, 0.01], \text{margin} = 0.4, \text{sigmoid} = \text{gaussian})$$

If $R_{\text{stage\_1}} = 1.0$, a large reward of 49 is added:

$$R_{\text{total}} = R_{\text{total}} + 49 \quad \text{if } R_{\text{stage\_1}} = 1.0$$

STAGE 2: MINIMIZE DISTANCE BETWEEN BALL AND TARGET

Let the position of the target be $\mathbf{p}_{\text{target}}$. Define the distance between the ball and the target as:

$$d_{\text{b}} = \|\mathbf{p}_{\text{ball}} - \mathbf{p}_{\text{target}}\|$$

The reward for minimizing this distance is:

$$R_{\text{stage\_2}} = \text{tolerance}(d_{\text{b}}, \text{bounds} = [0, 0.1], \text{margin} = 0.8, \text{sigmoid} = \text{gaussian})$$

TOTAL REWARD

The total reward is updated as:

$$R_{\text{total}} = R_{\text{total}} + R_{\text{stage\_1}} + R_{\text{stage\_2}}$$

If $R_{\text{stage\_2}} = 1.0$, indicating that the ball successfully reached the target, a large reward of 49 is added, and success is marked as:

$$R_{\text{total}} = R_{\text{total}} + 49, \quad \text{success} = \text{True}$$

Finally, the total reward is clipped to the range $[0, 100]$:

$$R_{\text{final}} = \min(\max(R_{\text{total}}, 0), 100)$$

SUCCESS CONDITION FOR SPARSE REWARD

The task is considered successful if:

$$\text{success} = \begin{cases} 1 & \text{if } R_{\text{stage\_2}} = 1.0 \\ 0 & \text{otherwise} \end{cases}$$

**Mini Golf.**

The reward function is composed of two stages:

STAGE 1: ROTATE THE TOOL

Let the rotation of the golf club be represented by $q_{\text{tool}}[2]$, which is the third component of its quaternion. The reward for rotating the tool is defined as:

$$R_{\text{rotate}} = \text{tolerance}(x = q_{\text{tool}}[2], \text{bounds} = [-0.4, -0.2], \text{margin} = 0.4, \text{sigmoid} = \text{gaussian})$$

STAGE 2: MOVE GOLF BALL TO TARGET

Let the position of the golf ball be $\mathbf{p}_{\text{ball}}$ and the position of the target be $\mathbf{p}_{\text{target}}$. Define the distance between them as:

$$d = \|\mathbf{p}_{\text{ball}} - \mathbf{p}_{\text{target}}\|$$

The reward for moving the golf ball to the target is:

$$R_{\text{move}} = \text{tolerance}(x = d, \text{bounds} = [0, 0.04], \text{margin} = 0.85, \text{sigmoid} = \text{gaussian})$$

### TOTAL REWARD

The total reward is the sum of the rewards from both stages:
$$R_{\text{total}} = R_{\text{rotate}} + R_{\text{move}}$$
If $R_{\text{move}} = 1.0$, indicating that the golf ball successfully reached the target, an additional bonus of 2.0 is added:
$$R_{\text{total}} = \begin{cases} R_{\text{total}} + 2.0 & \text{if } R_{\text{move}} = 1.0 \\ R_{\text{total}} & \text{otherwise} \end{cases}$$
Finally, the reward is clipped to the range $[0, 4]$:
$$R_{\text{final}} = \min(\max(R_{\text{total}}, 0), 4)$$

### SUCCESS CONDITION FOR SPARSE REWARD

The task is considered successful if:
$$\text{success} = \begin{cases} 1 & \text{if } R_{\text{move}} = 1.0 \\ 0 & \text{otherwise} \end{cases}$$

**Gather Cubes.**

The reward function consists of calculating rewards for three particle colors: red, green, and blue.

### REWARD FOR EACH PARTICLE

For each particle, the reward is calculated in two stages:

### STAGE 1: REACHING REWARD

Let the position of the tool be $\mathbf{p}_{\text{tool}}$, and the position of the particle (color $c$) be $\mathbf{p}_{\text{particle},c}$. Define the distance between them as:
$$d_{\text{reach},c} = \|\mathbf{p}_{\text{tool}} - \mathbf{p}_{\text{particle},c}\|$$
The reaching reward is given by:
$$R_{\text{reach},c} = \text{tolerance}\left(x = d_{\text{reach},c}, \text{bounds} = [0, 0.03175], \text{margin} = 0.12, \text{sigmoid} = \text{gaussian}\right)$$

### STAGE 2: MOVING TO BIN REWARD

Let the position of the bin target be $\mathbf{p}_{\text{target}}$. Define the distance between the particle (color $c$) and the bin target as:
$$d_{\text{bin},c} = \|\mathbf{p}_{\text{particle},c} - \mathbf{p}_{\text{target}}\|$$
The moving reward is given by:
$$R_{\text{move},c} = \text{tolerance}\left(x = d_{\text{bin},c}, \text{bounds} = [0, 0.075], \text{margin} = 0.1825, \text{sigmoid} = \text{gaussian}\right)$$

If the particle reaches the bin ($R_{\text{move},c} = 1.0$), an additional bonus of 2.0 is added:
$$R_{\text{particle},c} = R_{\text{reach},c} + R_{\text{move},c} + \begin{cases} 2.0 & \text{if } R_{\text{move},c} = 1.0 \\ 0 & \text{otherwise} \end{cases}$$
A success state is recorded for particle $c$:
$$\text{success}_c = \begin{cases} 1 & \text{if } R_{\text{move},c} = 1.0 \\ 0 & \text{otherwise} \end{cases}$$

### TOTAL REWARD

The total reward is the sum of the rewards for all particles:
$$R_{\text{total}} = \sum_{c \in \{\text{red, green, blue}\}} R_{\text{particle},c}$$

If all particles reach the bin ($\text{success}_{\text{red}} = \text{success}_{\text{green}} = \text{success}_{\text{blue}} = 1$), a large bonus of 5.0 is added:
$$R_{\text{total}} = \begin{cases} R_{\text{total}} + 5.0 & \text{if all particles succeed} \\ R_{\text{total}} & \text{otherwise} \end{cases}$$

### SUCCESS CONDITION FOR SPARSE REWARD

The task is considered successful if:

$$\text{success} = \begin{cases} 1 & \text{if all particles succeed} \\ 0 & \text{otherwise} \end{cases}$$

**Hammer Nail.**

The reward function consists of two main stages with an auxiliary reward in the first stage.

### STAGE 1: ALIGN MARKER WITH INITIAL NAIL TARGET

Let the position of the marker be $\mathbf{p}_{\text{marker}}$ and the position of the initial nail target be $\mathbf{p}_{\text{nail\_target\_initial}}$. Define the distance between them as:

$$d_{1\text{a}} = \|\mathbf{p}_{\text{marker}} - \mathbf{p}_{\text{nail\_target\_initial}}\|$$

The reward for minimizing this distance is:

$$R_{1\text{a}} = \text{tolerance}\,(x = d_{1\text{a}}, \text{bounds} = [0, 0.01], \text{margin} = 0.17, \text{sigmoid} = \text{gaussian})$$

### AUXILIARY REWARD: MAINTAIN SIMILAR HEIGHT

The height difference between the marker and the initial nail target is:

$$d_{1\text{b}} = p_{\text{marker},z} - p_{\text{nail\_target\_initial},z}$$

where $p_{\text{marker},z}$ and $p_{\text{nail\_target\_initial},z}$ are the $z$-coordinates of the marker and initial nail target, respectively. The auxiliary reward for minimizing this height difference is:

$$R_{1\text{b}} = \text{tolerance}\,(x = d_{1\text{b}}, \text{bounds} = [-0.01, 0.01], \text{margin} = 0.01, \text{sigmoid} = \text{gaussian})$$

### STAGE 2: ALIGN INITIAL NAIL TARGET WITH FINAL NAIL TARGET

Let the position of the final nail target be $\mathbf{p}_{\text{nail\_target\_final}}$. Define the distance between the initial and final nail targets as:

$$d_2 = \|\mathbf{p}_{\text{nail\_target\_initial}} - \mathbf{p}_{\text{nail\_target\_final}}\|$$

The reward for minimizing this distance is:

$$R_2 = \text{tolerance}\,(x = d_2, \text{bounds} = [0, 0.015], \text{margin} = 0.035, \text{sigmoid} = \text{gaussian})$$

### TOTAL REWARD

The total reward is the sum of the rewards from all stages:

$$R_{\text{total}} = R_{1\text{a}} + R_{1\text{b}} + R_2$$

### REWARD CLIPPING

The total reward is clipped to the range $[0, 3]$:

$$R_{\text{final}} = \min(\max(R_{\text{total}}, 0), 3)$$

### SUCCESS CONDITION FOR SPARSE REWARD

If $R_2 = 1.0$, indicating that the initial nail target successfully aligns with the final nail target, the task is considered successful:

$$\text{success} = \begin{cases} 1 & \text{if } R_2 = 1.0 \\ 0 & \text{otherwise} \end{cases}$$

### A.5 ENVIRONMENT RESET RANDOMIZATION

We apply position randomization to tools and objects in each environment, and include details for this below.

### A.5.1 Position Randomization for Tasks

Let the position randomization vector be denoted as:

$$\mathbf{r} = \begin{bmatrix} r_x \\ r_y \end{bmatrix}$$

where each component $r_x$ and $r_y$ is drawn randomly from a uniform distribution over the range $[-0.05, 0.05]$:

$$r_x, r_y \sim \mathcal{U}(-0.05, 0.05)$$

For each object (Tool, MoCap, and Particles) within the MuJoCo model, the position in the $x$ and $y$ directions is adjusted by adding this randomization:

$$\mathbf{p}_{\text{object}} = \mathbf{p}_{\text{object}} + \mathbf{r}$$

where:

$$\mathbf{p}_{\text{object}} = \begin{bmatrix} p_x \\ p_y \\ p_z \end{bmatrix}$$

is the original position of the object in 3D space. For all objects (Tool, MoCap, and Particles), the new position is updated as:

$$\mathbf{p}_{\text{object}}[:2] \leftarrow \mathbf{p}_{\text{object}}[:2] + \mathbf{r}$$

This ensures that only the $x$ and $y$ components of the position are modified, leaving the $z$-component unchanged.

- In Pouring, the same position randomization is applied to the tool/mocap and the particles inside the tool.
- In Scooping, we do not randomize the position of the particle – we only randomize the $(x, y)$ position of the tool/mocap.
- In Stacking, there are no particles, so only the position of the tool/mocap is randomized.
- In Scraping and Gathering, we apply 2 separate position randomizations to the tool/mocap and the objects the tool interacts with.
- In Ping Pong, we apply the same position randomization to $r_x$ (forward/backward placement) of the paddle and the ball.
- In Mini Golf, we apply the same position randomization to $r_y$ (horizontal placement) of the golf club and the ball. Additionally, we draw $r_y$ from $\mathcal{U}(-0.02, 0.02)$
- In Hammer Nail, we apply 2 separate position randomizations to the hammer/mocap $r_x$ (forward/backward placement) and the nail's $r_y$ (upward/downward placement).

