# OpenReview forum: "MuJoCo Manipulus: A Robot Learning Benchmark for Generalizable Tool Manipulation"
_ICLR.cc/2025/Conference — Submitted to ICLR 2025_

### Official Review · Reviewer_vJeu · 2024-10-26

**Soundness:** 2
**Presentation:** 3
**Contribution:** 2
**Rating:** 3
**Confidence:** 4

**Summary:**

The paper introduces MuJoCo Manipulus, a new benchmark for robotic tool manipulation. Built on the MuJoCo simulation platform, the benchmark includes 14 tasks and offers a user-friendly gym interface. The paper also provides baseline evaluations using SAC and PPO algorithms on the proposed tasks.

**Strengths:**

1. The paper is well-structured and presents its contributions clearly.
2. Tool manipulation is an important and timely topic in robotics, and the introduction of a new benchmark in this area is a valuable contribution.

**Weaknesses:**

1. A significant limitation of the benchmark is that the tasks are based on floating control points, rather than simulating direct interaction between the robot and the tool. This abstraction reduces the realism of the tasks and poses challenges for sim-to-real transfer, which is critical in robotic manipulation.
2. Despite the claim of offering a "diverse set of challenging tool manipulation tasks," the benchmark consists of only 14 tasks, with a significant portion focused on pouring. This narrow focus could limit the utility of the benchmark for evaluating a broader range of tool manipulation skills.

**Questions:**

I have no major technical questions, as the paper is relatively straightforward. However, I am unsure whether ICLR is the most appropriate venue for this work. While the benchmark provides meaningful contributions to the field of robotics, it lacks substantial algorithmic innovation or methodological novelty that would typically align with the scope of ICLR. A robotics-focused conference may offer a more fitting platform for this type of work.

---

> ### Author Response · Authors · 2024-12-01
> **Response to Reviewer vJeu**
>
> Thank you for your review of our work. Here, we address your concerns:
>
> **Pt. 1 “A significant limitation of the benchmark is that the tasks are based on floating control points, rather than simulating direct interaction between the robot and the tool. This abstraction reduces the realism of the tasks and poses challenges for sim-to-real transfer, which is critical in robotic manipulation.”**
>
> - Note, we provide the same response to this as for Reviewer P8eT, who raised a similar concern about our work:
> - Although there are no robots, this does not mean our benchmark is incomplete, or that it does not support Sim2Real research efforts.
>     - Direct control of tools is a common way to test tool manipulation, as demonstrated in related work such as https://robotic-tool-design.github.io/. The idea of using floating manipulators is not new either, as shown in the popular [Adroit Manipulation Platform](https://github.com/vikashplus/Adroit), and is a popular method for simulating 6-DoF motion without using an Inverse Kinematics library, such as [mink](https://github.com/kevinzakka/mink), which can slow down simulation speed since it adds additional CPU computation requirements to the task.
>     - Directly controlling a tool in our benchmark provides the user with (up to) 3 translational DoF and 3 rotational DoF, which added together, is on-par with the 6 DoF action space of a [UR5E Robot Arm from the MuJoCo Menagerie](https://github.com/google-deepmind/mujoco_menagerie/tree/main/universal_robots_ur5e). Assuming there is no grasping of the tool involved, and each tool’s canonical pose remains the same (as in our simulation), we can directly transfer the learned policies from our benchmark onto real robots.
>     - In comparison to the above “robotic tool design” work, we provide more tools and tasks, and include dense reward functions for training RL policies. Additionally, the constrained action spaces we introduce in our work are important and carefully designed to address the constraints and safety requirements of each tool. Please see our website, [mujoco-manipulus.github.io](http://mujoco-manipulus.github.io), which includes qualitative videos showing successful policy rollouts for each of our 16 tasks. There, you can see that the tool behavior exhibits natural motions and realistic physical simulation of all tools + objects. We believe this justifies the design of our constrained action spaces and reinforces the realistic behavior of tools in our benchmark.
>
> **Pt. 2 “Despite the claim of offering a "diverse set of challenging tool manipulation tasks," the benchmark consists of only 14 tasks, with a significant portion focused on pouring. This narrow focus could limit the utility of the benchmark for evaluating a broader range of tool manipulation skills.”**
>
> - We have updated our benchmark, which now includes 16 tool manipulation tasks and 14 tools from either MuJoCo or the TACO Dataset. More details about our tasks and tools are in our updated paper in Section 3.2. As mentioned in this section, the tools we provide can be broken down into 3 general categories: Kitchen Tools, Home Tools, and Sports Tools.
>     - Our “Kitchen Tools” category includes models for Bowl, Mug, Knife, Pan, Pot, Plate,
>     Spatula, Ladle, and Cup objects.
>     - Our “Home Tools” category includes models for a Brush, Hammer,
>     and Scooper.
>     - Our “Sports Tools” category includes models for a Ping-Pong Paddle and Golf Club.
> - We believe this is a strong representative set of tools to begin tool manipulation research with in a unified setting, and hope this addresses your specific concerns about whether our benchmark has diverse tool manipulation tasks.
> - Re: The difficulty of our tasks, we refer you to our paper’s updated Section 4.2, where we discuss the challenges in our more difficult tool manipulation tasks.
> - For additional tasks that are not included in our benchmark, as-is the case with other simulation benchmarks like ManiSkill, we will continue to update our benchmark with additional tasks. We will also open-source our benchmark, and encourage the community to make contributions with our task design guidelines to introduce new simulation tasks with our framework.
>
> **Pt. 3 Reviewer is concerned about whether ICLR is the most appropriate venue for this work**
>
> - ICLR supports a Datasets and Benchmarks “Primary Area” for paper submissions this year. We also highlight that robotic simulation works like ManiSkill2 have been accepted in prior ICLR conferences. Hence, we believe that our work has strong merit and relevance to ICLR.

---

### Official Review · Reviewer_DwWo · 2024-10-29

**Soundness:** 1
**Presentation:** 2
**Contribution:** 1
**Rating:** 3
**Confidence:** 4

**Summary:**

a Mujoco benchmark for policy learning including pouring, stacking, scooping, and various miscellaneous tasks, each with multiple tool options

**Strengths:**

This work introduces a MuJoCo simulator featuring a range of tasks, including pouring, stacking, scooping, and various miscellaneous tasks, each with multiple tool options.

The benchmark supports both state-based and image-based representations, compatible with standard reinforcement learning algorithms such as PPO and SAC.

**Weaknesses:**

The novelty of this benchmark appears limited. Many tasks resemble simplified, toy examples, with objects floating in the air rather than being manipulated by robotic arms. This raises concerns about the practical value of the benchmark for advancing robot learning.

The content is incomplete, with multiple paragraphs repeating similar ideas. While the state space for different tasks is described, there is no mention of the action space, reward function, or terminal states.

There is no comprehensive comparison with previous benchmarks, leaving the unique contributions of this benchmark unclear.

The number of objects or tools available for each task is limited, which restricts the potential for in-category object manipulation learning.

**Questions:**

1. Could different robot manipulators be added to the simulator? Most robotic arms are limited by their mechanical design and current motion planning algorithms, making it challenging to follow the trajectory of floating objects.

2. Could this benchmark support in-category object variations for each task? Adding this feature would greatly benefit the robot learning community.

3. A comparison table highlighting the differences and contributions of this benchmark compared to others would be very helpful.

---

> ### Author Response · Authors · 2024-12-01
> **Response to Reviewer DwWo (1/2)**
>
> Thank you for your review of our work. Here, we address your concerns:
>
> **Pt. 1 The novelty of our benchmark**
>
> - Please read through our detailed General Response to all reviewers, where we address this concern.
>
> **Pt. 2 “The content is incomplete, with multiple paragraphs repeating similar ideas. While the state space for different tasks is described, there is no mention of the action space, reward function, or terminal states.”**
>
> - Please read through our updated Sections 3, 4, and Appendix of our paper. These sections introduce new figures, results, and detailed discussions of {observation spaces, action spaces, environment resets, dense reward design, success conditions}.
> - Regarding terminal states, all tasks are 200 episodic steps, with the exception of `Ping Pong`, which is 50 episodic steps due to its short nature, and there are no terminal states which cause the episode to end early. All results are from episodes that last 200 or 50 episodic steps. We are concerned with ensuring that tools remain in a “success” state during the entire episode, and for this reason, we do not introduce early episode termination in our environments.
>
> **Pt. 3 “There is no comprehensive comparison with previous benchmarks, leaving the unique contributions of this benchmark unclear.”  / Pt. 4 “The number of objects or tools available for each task is limited, which restricts the potential for in-category object manipulation learning.”**
>
> - We added Table 1 + Appendix Section A.1 to comprehensively compare our work with existing works. In Sections 2 and 3 of our updated paper, we discuss the merits of our work over existing works too.
> - Related to Pt. 4, {Section 3, Table 1, Appendix Section A.1} address the in-category object variants for each of our tasks, as well as the tool skills that can be learned in our benchmark compared to existing benchmarks. There are additional details we provide, such as how we have dense rewards for all our tasks, and a diverse set of tools compared to existing works.
>
> **Q1 Could different robot manipulators be added to the simulator? Most robotic arms are limited by their mechanical design and current motion planning algorithms, making it challenging to follow the trajectory of floating objects.**
>
> - Yes, we can add different robot manipulators to our simulator, and plan to do this in the near future.
> - We repeat the following content from our response to Reviewer P8eT:
>     - Although there are no robots, this does not mean our benchmark is incomplete, or that it does not support Sim2Real research efforts.
>         - Direct control of tools is a common way to test tool manipulation, as demonstrated in related work such as https://robotic-tool-design.github.io/. The idea of using floating manipulators is not new either, as shown in the popular [Adroit Manipulation Platform](https://github.com/vikashplus/Adroit), and is a popular method for simulating 6-DoF motion without using an Inverse Kinematics library, such as [mink](https://github.com/kevinzakka/mink), which can slow down simulation speed since it adds additional CPU computation requirements to the task.
>         - Directly controlling a tool in our benchmark provides the user with (up to) 3 translational DoF and 3 rotational DoF, which added together, is on-par with the 6 DoF action space of a [UR5E Robot Arm from the MuJoCo Menagerie](https://github.com/google-deepmind/mujoco_menagerie/tree/main/universal_robots_ur5e). Assuming there is no grasping of the tool involved, and each tool’s canonical pose remains the same (as in our simulation), we can directly transfer the learned policies from our benchmark onto real robots.
>         - In comparison to the above “robotic tool design” work, we provide more tools and tasks, and include dense reward functions for training RL policies. Additionally, the constrained action spaces we introduce in our work are important and carefully designed to address the constraints and safety requirements of each tool. Please see our website, [mujoco-manipulus.github.io](http://mujoco-manipulus.github.io), which includes qualitative videos showing successful policy rollouts for each of our 16 tasks. There, you can see that the tool behavior exhibits natural motions and realistic physical simulation of all tools + objects. We believe this justifies the design of our constrained action spaces and reinforces the realistic behavior of tools in our benchmark.

---

> > ### Author Response · Authors · 2024-12-01
> > **Response to Reviewer DwWo (2/2)**
> >
> > **Q2 Could this benchmark support in-category object variations for each task? Adding this feature would greatly benefit the robot learning community.**
> >
> > - Yes, and it already supports in-category object variants for each task. Please see Section 3.2 of our updated paper, where we show object variants for each task category. For example, our Pouring task supports a Cup, Mug, Pan, Pot, Bowl, and Plate. Our framework is flexible, and if a user follows the steps in our design pipeline shown in Figure 2 of our paper, more in-category object variants can be added to a task if one desires.
> >
> > **Q3 A comparison table highlighting the differences and contributions of this benchmark compared to others would be very helpful.**
> >
> > - See Table 1 in our paper + Appendix Section A.1 for this. Thank you for recommending this addition to our work.

---

### Official Review · Reviewer_AaoX · 2024-11-01

**Soundness:** 2
**Presentation:** 2
**Contribution:** 2
**Rating:** 3
**Confidence:** 5

**Summary:**

This paper introduced a MuJoCo-based simulation framework named "MuJoCo Manipulus," which features tool-use tasks such as pouring, stacking, scooping, gathering, etc. In these benchmarking tasks, an "idealized" agent manipulates a free-floating tool to physically interact with objects and accomplish manipulation goals. Specifically, the initial release of this benchmark includes 12 tools spanning two categories: Kitchen Tools and Home Tools, which were either hand-designed or drawn from existing asset libraries (e.g., TACO and MuJoCo). The authors highlighted several engineering features offered by their framework, including support for end-to-end RL with the MuJoCo physics engine, integration with the Gymnasium interface, and state/vision-based observations. As an initial attempt toward solving these tasks, they evaluated two popular deep RL algorithms, PPO (on-policy) and SAC (off-policy), with both state-based and vision-based observations. These baseline methods achieved mixed results in these tasks, suggesting room for future development.

**Strengths:**

- **Problem motivation.** Tool use is an important family of robot manipulation problems to study, as it represents a high level of physical intelligence manifested in intelligent species and requires multi-step, contact-rich interactions with multiple objects. Building systematic benchmarks for robotic tool manipulation would facilitate reproducible research in this problem domain.
- **User-friendly design.** The proposed benchmark was built on the MuJoCo simulation engine. Its designs prioritized flexibility and ease of use, with the standardized Gymnasium APIs and support for state-based/vision-based observations for end-to-end reinforcement learning. These designs could reduce the barrier of entry for researchers to leverage this simulator for research.
- **Baseline evaluations.** Experiments with standard RL algorithms, including PPO and SAC, presented a baseline view of existing RL methods' strengths and limitations. These preliminary results provide insight into the major challenges and motivate potential avenues for future work.

**Weaknesses:**

While the benchmark is well-motivated and carefully designed, its current version bears major weaknesses:

- Although the authors motivated this benchmark for robot learning and manipulation research, this paper did not use realistic robot models or controllers. Instead, it used an idealized "point mass" robot with simple Cartesian action spaces. This design severely abstracts away the challenges of contact-rich robot manipulation problems in the real world. Compared to existing manipulation benchmarks, such as ManiSkill2 (Gu et al., 2023) and Robosuite (Zhu et al., 2020), which aimed to model robot dynamics faithfully, the simplified robot model could hinder the ability to deploy the learned models on physical robots and thus limit its impact in advancing robot learning research.
- The authors motivated the benchmark as a "wider-scale tool manipulation benchmark with substantially more tasks." However, the current offering of the tasks is fairly limiting compared to several existing simulation frameworks for robot navigation and manipulation (Nasiriany et al., 2024; Szot et al., 2021; Li et al., 2022), which provides 100-1000 tasks in large-scale home environments. In contrast, the proposed benchmark only provides 14 simplified tasks in a restricted tabletop workspace.
- The authors highlighted the "Unified API for Reinforcement Learning with MuJoCo" as the main feature of the benchmark and spent a great amount of text in Sec 3 explaining these design choices. However, these features seem to be directly inherited from MuJoCo and its derived open-source tools/software, such as Gymsium. MuJoCo provides native support for state-based/vision-based observations, which has been commonly used in existing frameworks such as Robosuite (Zhu et al., 2020). Therefore, it remains unclear what technical challenges the authors have tackled to construct this benchmark and what are unique designs of this benchmark that have not been done in prior work.
- The experiments of this paper focus on evaluating off-the-shelf RL methods instead of innovating new ones. More qualitative analyses and ablative studies should be done to understand the limitations of these methods and provide insight into how these algorithms could be improved to better tackle these tasks. The mixed success rates also indicate the challenges of defining informative (dense) reward functions without extensive reward engineering. More reward design techniques could be explored to improve the quality of these tasks for RL purposes.

**Questions:**

In addition to addressing my comments in the Weaknesses section above, I have the following questions:
- I would like to hear the authors' explanation of the technical contributions of this work in comparison to existing MuJoCo-based simulations and tools. The APIs, simulation speed, and environments do not seem particularly novel. Can you please clarify how this effort differentiates (improves upon) robot learning simulations, especially the ones built with MuJoCo?
- Why would different tasks come with different action spaces, from three to five DoFs? For example, the Pouring tasks have 4D action spaces, while the Stacking tasks have 3D ones. Would it be beneficial to unify the actions across these tasks to enable multi-task RL research with the benchmark?
- Could you comment on the potential of transferring the model trained on MuJoCo Manipulus to a real robot? How could you bridge the embodiment gap between the idealized robot used in this work and the kinematic designs of real robots?
- The RL results reported that "the PPO+RGB and SAC+RGB methods were often better than PPO+State and SAC+State." This result is interesting and a bit counterintuitive, as visual RL is usually more expensive to train due to the high-dimensional visual observations as input to the policy. Could you provide more insight why we see this result?

---

> ### Author Response · Authors · 2024-12-01
> **Response to Reviewer AaoX (1/3)**
>
> Thank you for your review of our work. Here, we address each of the Weaknesses you mentioned:
>
> **Pt. 1 Although the authors motivated this benchmark for robot learning and manipulation research, this paper did not use realistic robot models or controllers. Instead, it used an idealized "point mass" robot with simple Cartesian action spaces. This design severely abstracts away the challenges of contact-rich robot manipulation problems in the real world. Compared to existing manipulation benchmarks, such as ManiSkill2 (Gu et al., 2023) and Robosuite (Zhu et al., 2020), which aimed to model robot dynamics faithfully, the simplified robot model could hinder the ability to deploy the learned models on physical robots and thus limit its impact in advancing robot learning research.**
>
> - We also discussed the reasoning behind our Floating Tool Control model with other reviewers who shared this concern, and provide the following response:
>     - Although there are no robots, this does not mean our benchmark is incomplete, or that it does not support Sim2Real research efforts.
>     - Direct control of tools is a common way to test tool manipulation, as demonstrated in related work such as https://robotic-tool-design.github.io/. The idea of using floating manipulators is not new either, as shown in the popular [Adroit Manipulation Platform](https://github.com/vikashplus/Adroit), and is a popular method for simulating 6-DoF motion without using an Inverse Kinematics library, such as [mink](https://github.com/kevinzakka/mink), which can slow down simulation speed since it adds additional CPU computation requirements to the task.
>     - Directly controlling a tool in our benchmark provides the user with (up to) 3 translational DoF and 3 rotational DoF, which added together, is on-par with the 6 DoF action space of a [UR5E Robot Arm from the MuJoCo Menagerie](https://github.com/google-deepmind/mujoco_menagerie/tree/main/universal_robots_ur5e). Assuming there is no grasping of the tool involved, and each tool’s canonical pose remains the same (as in our simulation), we can directly transfer the learned policies from our benchmark onto real robots.
>     - In comparison to the above “robotic tool design” work, we provide more tools and tasks, and include dense reward functions for training RL policies. Additionally, the constrained action spaces we introduce in our work are important and carefully designed to address the constraints and safety requirements of each tool. Please see our website, [mujoco-manipulus.github.io](http://mujoco-manipulus.github.io), which includes qualitative videos showing successful policy rollouts for each of our 16 tasks. There, you can see that the tool behavior exhibits natural motions and realistic physical simulation of all tools + objects. We believe this justifies the design of our constrained action spaces and reinforces the realistic behavior of tools in our benchmark.
>
>
> **Pt. 2 The authors motivated the benchmark as a "wider-scale tool manipulation benchmark with substantially more tasks." However, the current offering of the tasks is fairly limiting compared to several existing simulation frameworks for robot navigation and manipulation (Nasiriany et al., 2024; Szot et al., 2021; Li et al., 2022), which provides 100-1000 tasks in large-scale home environments. In contrast, the proposed benchmark only provides 14 simplified tasks in a restricted tabletop workspace.**
>
> - Robot Navigation and Embodied AI works, which the reviewer has mentioned, are focused on high-level planning for robots. Our benchmark is for low-level control of tools and robots, with an emphasis on RL algorithms for learning low-level control. Similar benchmarks to our work are included in Table 1 of our paper and discussed in Section 2 of our paper, and we provide a comparison our Benchmark’s Tool Use Skills compared to other benchmarks in Appendix Section A.1. We also include details of our low-level tool control tasks in Section 3 of our work.
> - For manipulation benchmarks such as RLBench, and Embodied AI works that include Navigation and Mobile Manipulation, they provide no dense rewards for guiding RL policy learning. RL algorithms are typically applied to low-level control problems like those found in our work and related benchmarks such as ManiSkill and Robosuite. In the case of RLBench, while they provide many manipulation tasks, they only provide sparse rewards, which are hard to learn low-level control policies with using plain model-free RL algorithms. We’d like to mention that our benchmark jointly includes dense rewards and sparse rewards for all tasks.

---

> > ### Author Response · Authors · 2024-12-01
> > **Response to Reviewer AaoX (2/3)**
> >
> > **Pt. 3 The authors highlighted the "Unified API for Reinforcement Learning with MuJoCo" as the main feature of the benchmark and spent a great amount of text in Sec 3 explaining these design choices. However, these features seem to be directly inherited from MuJoCo and its derived open-source tools/software, such as Gymsium. MuJoCo provides native support for state-based/vision-based observations, which has been commonly used in existing frameworks such as Robosuite (Zhu et al., 2020). Therefore, it remains unclear what technical challenges the authors have tackled to construct this benchmark and what are unique designs of this benchmark that have not been done in prior work.**
> >
> > - Please see Figure 2 and our updated Section 3, Section 4, and Appendix of our paper for detailed discussion of our benchmark’s simulation task design pipeline, discussion of each task category, results, and environment details {observation spaces, action spaces, environment resets, success conditions, success/failure visualizations, etc.}.
> > - We’d like to clarify to the reviewer that there is no standard API for creating RL tasks with MuJoCo. MuJoCo is a Physics Simulation Engine, which happens to support Robotics and RL research because it can simulate contact-rich environments well. Benchmarks like Robosuite and Meta-World are built on top of MuJoCo, but introduced their own API to do RL on top of MuJoCo. Both of these benchmarks were published in 2019, before MuJoCo was open-source, and still have old conventions of MuJoCo in their codebase.
> > - Our codebase uses MuJoCo 3.0+, the latest version of MuJoCo, and the new Farama Gymnasium API for RL. To our knowledge, there is no benchmark that has all the modern features of MuJoCo and Gymnasium like our work, all while still following an elegant design scheme like our work, and includes easy installation and end-to-end usage on a Mac or Linux machine. All of our environments are defined in single files, following a similar design philosophy to CleanRL, except it’s our goal to provide elegant single-file RL *environment* implementations instead of RL *algorithm* implementations. We see this as a huge benefit to RL practitioners and novices who want to design their own environments, and are hopeful that this will lower the barrier of entry for those interested in starting RL research.
> >
> > **Pt. 4 The experiments of this paper focus on evaluating off-the-shelf RL methods instead of innovating new ones. More qualitative analyses and ablative studies should be done to understand the limitations of these methods and provide insight into how these algorithms could be improved to better tackle these tasks. The mixed success rates also indicate the challenges of defining informative (dense) reward functions without extensive reward engineering. More reward design techniques could be explored to improve the quality of these tasks for RL purposes.**
> >
> > - We have provided qualitative visualizations of the best-performing policy for each task on our website, [mujoco-manipulus.github.io](http://mujoco-manipulus.github.io). We also provide Figure 9 in our paper, which includes the Success Rate curves for all 16 of our tool manipulation tasks using 3 well-established model-free RL algorithms: CrossQ, SAC, and PPO. With dense reward engineering, which we provide extensive details of in Appendix Section A.4, we were able to learn reasonable control policies for each of our tasks. There is still room for improvement though, and we include discussion for *where* we can improve on harder tasks in our General Response to reviewers in Pt. 1. Specifically, areas of improvement include the **Sample-Efficiency of RL Algorithms, Off-Policy RL methods for Short-Horizon Dynamic Tasks,** and **Learning Reward Functions.**
> >
> > **Q1 I would like to hear the authors' explanation of the technical contributions of this work in comparison to existing MuJoCo-based simulations and tools. The APIs, simulation speed, and environments do not seem particularly novel. Can you please clarify how this effort differentiates (improves upon) robot learning simulations, especially the ones built with MuJoCo?**
> >
> > - This is generally answered in our Pt. 3 response. We also provide useful metrics, such as simulation speed of our benchmark, in Section 4.2 of our updated paper. In short, our benchmark is reasonably fast, with 100K training steps taking 10 minutes of walltime, and 300K training steps taking 30 minutes of walltime.

---

> > > ### Author Response · Authors · 2024-12-01
> > > **Response to Reviewer AaoX (3/3)**
> > >
> > > **Q2 Why would different tasks come with different action spaces, from three to five DoFs? For example, the Pouring tasks have 4D action spaces, while the Stacking tasks have 3D ones. Would it be beneficial to unify the actions across these tasks to enable multi-task RL research with the benchmark?**
> > >
> > > - In short, different tools have different constraints (to ensure they are safely being used) and different behaviors (see our videos at [mujoco-manipulus.github.io](http://mujoco-manipulus.github.io)), which motivates the need for each set of tools having a different set of constraints. It is indeed possible to make all the tasks 6 DoF, but we restrict the number of DoFs mainly to enable off-the-shelf RL algorithms to make reasonable learning progress, and to prevent possibly unsafe behaviors from being learned with full 6 DoF control. Otherwise, even if a task is being solved somehow in simulation, it may not be safe for Sim2Real deployment (e.g. in our `ScrapeKnife` task, we want the knife to move slowly and within a confined space, away from harming any human users).
> > >
> > > **Q3 Could you comment on the potential of transferring the model trained on MuJoCo Manipulus to a real robot? How could you bridge the embodiment gap between the idealized robot used in this work and the kinematic designs of real robots?**
> > >
> > > - This is generally answered in our Pt. 1 response.
> > >
> > > **Q4 The RL results reported that "the PPO+RGB and SAC+RGB methods were often better than PPO+State and SAC+State." This result is interesting and a bit counterintuitive, as visual RL is usually more expensive to train due to the high-dimensional visual observations as input to the policy. Could you provide more insight why we see this result?**
> > >
> > > - This is addressed in our General Response to all reviewers, specifically in Pt. 3 Part C. “**Removed RGB Training Results”**. In short, our benchmark focuses on whether we can learn low-level tool manipulation policies with RL, and we use state information from MuJoCo as inputs to our policies. This is because policies trained with state inputs can often represent the upper bound of performance for any of our 3 model-free RL algorithm baselines.
> > > - We leave research on Visual RL for tool manipulation for future work that uses our benchmark. To clarify, our benchmark supports state-only, RGB-only, and state+RGB inputs for policies.

---

> > > > ### Comment · Reviewer_AaoX · 2024-12-03
> > > >
> > > > I appreciate the authors' efforts in preparing the detailed responses to my questions. The rebuttal clarified some of my questions, but unfortunately, the primary issues about the lack of realistic robot models/controllers and the limited task diversity remain unresolved. I also agree with Reviewer P8eT's point that integrating robots into this benchmark could make it more relevant and appealing to the embodied AI/robotics research communities. In addition, it would substantially benefit the quality of the paper if the authors further increased the variability of the tasks and demonstrated new algorithmic insights through experimentation with this benchmark. Considering these, I will retain my original rating for this paper.

---

### Official Review · Reviewer_eoGS · 2024-11-02

**Soundness:** 3
**Presentation:** 4
**Contribution:** 2
**Rating:** 5
**Confidence:** 4

**Summary:**

This paper presents Mujoco Manipulus, a tool use benchmark for RL research.  This benchmark is built on top of Mujoco and gynmasium, and supports a range of tasks and tools such as scooping, stacking and gathering.  The authors evaluate PPO and SAC on these tasks using both vision and state information, demonstrating that these tasks are nontrivial to solve.

**Strengths:**

+ Developing new benchmarks for robotic skills and manipulation is an important contribution to the community.
+ The paper is well presented and easy to understand.
+ The authors present a clear picture of related work.

**Weaknesses:**

- The size of the benchmark is relatively small.  There are only a few tasks and objects available.
- The complexity of these tasks is relatively low.  The agent has direct control over the tools (non-dexterous) and requires only a few degrees of freedom to move them.
- The realism is fairly limited, while at the same time the simulator uses Mujoco on the CPU.  Most recent robotics benchmarks are moving toward either greater realism or very-high-throughput simulation on the GPU, and so this benchmark, which lacks both, feels somewhat behind the Pareto frontier.
- It's not clear how challenging this benchmark is.  The authors run PPO and SAC on these tasks and show that they struggle to solve them out of the box.  In many robotics applications though, a large degree of reward shaping or learning from demonstrations is frequently necessary to solve real world problems, and it's not clear why those approaches wouldn't work here as well.
- A more thorough comparison against relevant tasks from Maniskill2 and Robosuite is necessary to establish the utility/necessity of this new benchmark.  Those two settings contain some tool use, are the tasks here harder than those?  More comprehensive?  Given the simplicity of the models presented here, it seems like the others should be more realistic.  What new features does this benchmark include that are not covered elsewhere?  If the only contribution here is to package together certain types of tasks that exist across many other benchmarks, that could still be useful to the community, but more work needs to be done to establish the added value of the dataset relative to what's already out there.
- Along the same lines, in a more general sense it is not clear what utility the broader community gets from this new benchmark.  Given the limited realism, this doesn't seem like it advances our practical ability to train real-world robots (sim-to-real does not seem to be much of a consideration here).  Given the small size, it doesn't add much to the already substantial physically grounded simulation benchmarks for RL.  The authors motivate this benchmark by arguing that there is not a dedicated tool-use benchmark for robot manipulation, but there are several that contain tool use as a subset, and it's not clear what this new one adds to them.

**Questions:**

The authors mention Maniskill2 and Robosuite, but say that only a portion of those tasks cover tool use/manipulation.  How would the current benchmark compare in size/complexity/realism with a subset of the Maniskill2 and Robosuite tasks that only cover tool use/manipulation?

---

> ### Author Response · Authors · 2024-12-01
> **Response to Reviewer eoGS (1/2)**
>
> Thank you for your review of our work. Here, we address your concerns:
>
> **Pt. 1 The size of the benchmark is relatively small**
>
> - The updated size of our benchmark is 16 tasks, and we provide 14 tools in our benchmark. For comparison to other benchmarks, please see Table 1 and Appendix Section A.1 in our updated paper. Our benchmark provides Dense Rewards for all tasks, a critical feature for guiding RL policy learning. Compared to existing benchmarks such as ManiSkill and Robosuite, which also provide Dense Rewards, we are on-par with the size of ManiSkill 3’s tabletop manipulation task suite (16 tasks) and larger than Robosuite (9 tasks) while offering more tool interaction options than both benchmarks.
>
> **Pt. 2 The complexity of these tasks is relatively low. The agent has direct control over the tools (non-dexterous) and requires only a few degrees of freedom to move them.**
>
> - We discuss our more challenging tool manipulation tasks in Section 4.2 of our updated paper, and in our General Response to all reviewers. We also discuss the reasoning behind our Floating Tool Control model with other reviewers who shared this concern, and provide the following response:
>     - Although there are no robots, this does not mean our benchmark is incomplete, or that it does not support Sim2Real research efforts.
>     - Direct control of tools is a common way to test tool manipulation, as demonstrated in related work such as https://robotic-tool-design.github.io/. The idea of using floating manipulators is not new either, as shown in the popular [Adroit Manipulation Platform](https://github.com/vikashplus/Adroit), and is a popular method for simulating 6-DoF motion without using an Inverse Kinematics library, such as [mink](https://github.com/kevinzakka/mink), which can slow down simulation speed since it adds additional CPU computation requirements to the task.
>     - Directly controlling a tool in our benchmark provides the user with (up to) 3 translational DoF and 3 rotational DoF, which added together, is on-par with the 6 DoF action space of a [UR5E Robot Arm from the MuJoCo Menagerie](https://github.com/google-deepmind/mujoco_menagerie/tree/main/universal_robots_ur5e). Assuming there is no grasping of the tool involved, and each tool’s canonical pose remains the same (as in our simulation), we can directly transfer the learned policies from our benchmark onto real robots.
>     - In comparison to the above “robotic tool design” work, we provide more tools and tasks, and include dense reward functions for training RL policies. Additionally, the constrained action spaces we introduce in our work are important and carefully designed to address the constraints and safety requirements of each tool. Please see our website, [mujoco-manipulus.github.io](http://mujoco-manipulus.github.io), which includes qualitative videos showing successful policy rollouts for each of our 16 tasks. There, you can see that the tool behavior exhibits natural motions and realistic physical simulation of all tools + objects. We believe this justifies the design of our constrained action spaces and reinforces the realistic behavior of tools in our benchmark.
>
> **Pt. 3 The realism is fairly limited, while at the same time the simulator uses Mujoco on the CPU. Most recent robotics benchmarks are moving toward either greater realism or very-high-throughput simulation on the GPU, and so this benchmark, which lacks both, feels somewhat behind the Pareto frontier.**
>
> - Please see Point #6, “**Is Our Benchmark on the Pareto Frontier of Robotics?”**, in our General Response to all reviewers. We believe this is an important topic to mention to all reviewers, and thank the reviewer for bringing this concern to our attention.
>
> **Pt. 4 It's not clear how challenging this benchmark is. The authors run PPO and SAC on these tasks and show that they struggle to solve them out of the box. In many robotics applications though, a large degree of reward shaping or learning from demonstrations is frequently necessary to solve real world problems, and it's not clear why those approaches wouldn't work here as well.**
>
> - Please see our updated Figure 9 in our paper, which shows the Success Rate curves for all 16 of our tasks. In short, we applied 3 model-free RL baselines to our tasks, and were able to learn meaningful behaviors across all tasks with ≥ 1 baseline algorithm. For qualitative visualizations of the best-performing policies for each task, please visit [mujoco-manipulus.github.io](http://mujoco-manipulus.github.io). The results we achieve are thanks to additional Reward Engineering, which we include extensive discussion of in our new Appendix Section A.4.

---

> > ### Author Response · Authors · 2024-12-01
> > **Response to Reviewer eoGS (2/2)**
> >
> > **Pt. 5 A more thorough comparison against relevant tasks from Maniskill2 and Robosuite is necessary to establish the utility/necessity of this new benchmark. Those two settings contain some tool use, are the tasks here harder than those? More comprehensive? Given the simplicity of the models presented here, it seems like the others should be more realistic. What new features does this benchmark include that are not covered elsewhere? If the only contribution here is to package together certain types of tasks that exist across many other benchmarks, that could still be useful to the community, but more work needs to be done to establish the added value of the dataset relative to what's already out there.**
> >
> > - More thorough comparison is provided against related benchmarks in Table 1 and Appendix Section A.1 of our updated paper. We also include discussion of related works in Section 2 of our paper, and discuss the merits of our benchmark over existing benchmarks in Section 3 of our paper.
> > - While we cannot make one-to-one comparisons between benchmarks, we can confirm that our benchmark includes challenging tool manipulation tasks and a more comprehensive set of tool manipulation tasks, and includes more of these tasks in a unified setting than other benchmarks we compare against in our paper. This includes Robosuite and ManiSkill2.
> > - The new features of our benchmark can be simplified to two things:
> >     1. **An elegant, battle-tested pipeline for designing MuJoCo-Gymnasium RL tasks and training RL policies end-to-end**. A complete overview of our framework is provided in Figure 2 of our updated paper, and discussed in Section 3.1 of our paper.
> >     2. **The tasks**. Our unified offering of 16 tool manipulation tasks will accelerate research in this subfield of robotic manipulation research, and empower broader research in RL, Vision, and Robotics.
> >
> > **Pt. 6 Along the same lines, in a more general sense it is not clear what utility the broader community gets from this new benchmark. Given the limited realism, this doesn't seem like it advances our practical ability to train real-world robots (sim-to-real does not seem to be much of a consideration here). Given the small size, it doesn't add much to the already substantial physically grounded simulation benchmarks for RL. The authors motivate this benchmark by arguing that there is not a dedicated tool-use benchmark for robot manipulation, but there are several that contain tool use as a subset, and it's not clear what this new one adds to them.**
> >
> > - We address the Sim2Real concern above in our Pt. 2 response.
> > - Re: “**it's not clear what this new one adds to them”, t**his point is generally answered by Pt. 5.
> >
> > **Q1 The authors mention ManiSkill2 and Robosuite, but say that only a portion of those tasks cover tool use/manipulation. How would the current benchmark compare in size/complexity/realism with a subset of the Maniskill2 and Robosuite tasks that only cover tool use/manipulation?**
> >
> > - This point is generally answered by Pt. 5.

---

### Official Review · Reviewer_P8eT · 2024-11-05

**Soundness:** 2
**Presentation:** 2
**Contribution:** 1
**Rating:** 3
**Confidence:** 5

**Summary:**

The paper presents a benchmark designed for robotic tool manipulation using the MuJoCo physics engine. It offers a diverse set of 14 tool manipulation tasks across categories such as pouring, stacking, scooping, gathering, hammering, and scraping. Each task features specific tools (e.g., cup, ladle, hammer) and allows for both state-based and vision-based learning. The benchmark integrates with Gymnasium API and popular reinforcement learning (RL) libraries, making it easily used by the research community.

**Strengths:**

1. The paper is well-written with clear images.
2. It presents convenient APIs and environments, built on the MuJoCo engine, incorporating various tools and objects.
3. Several commonly used RL algorithms were tested.

**Weaknesses:**

1. The benchmark lacks completeness; as a robot learning environment, it does not include any robots, and the tasks and action spaces are very simple.
2. Novelty and contribution are severely concerned - it feels more like an engineering project. Essentially, it imports objects into MuJoCo, designs tasks around them, and tests some off-the-shelf algorithms.
3. The action space appears to involve directly transforming the pose of the objects (tools) rather than applying interaction with them, which raises severe concerns about whether true physical simulation is involved.
4. Current SOTA simulation benchmarks could support the tasks presented in this paper by simply importing the relevant 3D objects.

In summary, this is a reasonable project but lacks completeness (at the very least, a robot should be included as a robot learning environment) and novelty, so I recommend rejecting it.

**Questions:**

No additional questions.

---

> ### Author Response · Authors · 2024-12-01
> **Response to Reviewer P8eT**
>
> Thank you for your review of our work. Here, we address each of the Weaknesses you mentioned:
>
> **Pt. 1 The benchmark lacks completeness / Pt. 3 Is True Physical Simulation involved?**
>
> - We are the first simulation benchmark to present a complete packaging of 14 tools and 16 tool manipulation tasks, all with an easy-to-use Reinforcement Learning interface for robotics and RL practitioners.
> - Although there are no robots, this does not mean our benchmark is incomplete, or that it does not support Sim2Real research efforts.
>     - Direct control of tools is a common way to test tool manipulation, as demonstrated in related work such as https://robotic-tool-design.github.io/. The idea of using floating manipulators is not new either, as shown in the popular [Adroit Manipulation Platform](https://github.com/vikashplus/Adroit), and is a popular method for simulating 6-DoF motion without using an Inverse Kinematics library, such as [mink](https://github.com/kevinzakka/mink), which can slow down simulation speed since it adds additional CPU computation requirements to the task.
>     - Directly controlling a tool in our benchmark provides the user with (up to) 3 translational DoF and 3 rotational DoF, which added together, is on-par with the 6 DoF action space of a [UR5E Robot Arm from the MuJoCo Menagerie](https://github.com/google-deepmind/mujoco_menagerie/tree/main/universal_robots_ur5e). Assuming there is no grasping of the tool involved, and each tool’s canonical pose remains the same (as in our simulation), we can directly transfer the learned policies from our benchmark onto real robots.
>     - In comparison to the above “robotic tool design” work, we provide more tools and tasks, and include dense reward functions for training RL policies. Additionally, the constrained action spaces we introduce in our work are important and carefully designed to address the constraints and safety requirements of each tool. Please see our website, [mujoco-manipulus.github.io](http://mujoco-manipulus.github.io), which includes qualitative videos showing successful policy rollouts for each of our 16 tasks. There, you can see that the tool behavior exhibits natural motions and realistic physical simulation of all tools + objects. We believe this justifies the design of our constrained action spaces and reinforces the realistic behavior of tools in our benchmark.
>
> **Pt. 2 Novelty and Contribution are severely concerned / Pt. 4 Current Simulation Benchmarks can support these tasks by simply importing the relevant 3D objects**
>
> - Please see Figure 2 in our updated paper, and read through our renovated Section 3 of our paper for a detailed discussion of our benchmark’s design pipeline. In short, it is not as simple as importing the relevant 3D objects into the simulator. We additionally have added an Appendix with Reward Engineering details, and a new Table 1 + Appendix Section A.1 in our paper with a comparison of our benchmark to existing simulation benchmarks.
> - We also hope that the General Response to all reviewers addresses your concerns about the novelty of our work, and why we chose MuJoCo instead of a different simulation backend (see Point #6 in our General Response to all reviewers).

---

> ### Comment · Reviewer_P8eT · 2024-12-01
>
> Thank you for the clarification. I appreciate the visualization (the website) and the additional results. However, I still encourage the authors to integrate robots into the benchmark. Even the listed related works in this rebuttal have either a full Franka or a dexterous hand, not a floating point like this paper.
>
> For policy learning, it's possible to use a 6DoF point to learn trajectory, but for a robotic benchmark paper, it must contain a robotic embodiment (at least an end-effector). The sim-to-real gap is not that trivial, from a robotic side, even a photo-realistic simulator with accurate models of robots can cause problems when transferring from sim to real.
>
> Therefore, a possible way to improve this paper is to import robots, or to consider it as a dataset paper providing tool using demos. An exception to this is that you are trying to tackle super-hard problems, such as simulating soft object manipulation, where you can focus on the object itself, not the robotic side. But obviously, tools are rigid objects, the difficulties is in robotic grasping and manipulation. I am afraid that it cannot be accepted by robotics-related conferences before importing robots (at least EE) into the benchmark.
>
> Please consider improving it with these suggestions.

---

### Author Response · Authors · 2024-12-01
**General Response to Reviewers (1/3)**

Dear Reviewers,

Thank you for your time reviewing our work, and for providing detailed feedback. We appreciate each of your reviews, and have worked hard to address several of your concerns. Below, we provide a summary of key updates to our work:

1. **Updated Results w/ New Dense Reward Functions**
    - We agree with reviewers that reward engineering is highly important for our Reinforcement Learning benchmark, and have spent time improving our Reward Functions for all tasks. In the updated draft of our work, please see Figure 9, where you can find Success Rate curves for the 16 Tool Manipulation tasks in our benchmark. These results showcase the effectiveness of our new rewards, and indicate that there is still a wide margin of improvement to be made in our more challenging tasks.
        1. Our benchmark’s harder tasks include `Scoop Particle`, `Scoop Cube`, `Stack Plates` , `Stack Bowls`, and `Ping Pong`. We provide a table below, showing the Final Average Success Rates across 5 seeds for our 5 most-challenging tasks:


            | Task ID | # of Training Steps | CrossQ | SAC | PPO | Num Steps per Episode | Total Episodes of Experience for Training |
            | --- | --- | --- | --- | --- | --- | --- |
            | Scoop Particle | 300,000 | **98.4** | 43.6 | 0 | 200 | 1500 |
            | Scoop Cube | 300,000 | **98.4** | 18.4 | 0 | 200 | 1500 |
            | Stack Plates | 300,000 | **76.8** | 0 | 0 | 200 | 1500 |
            | Stack Bowls | 300,000 | **34.4** | 0 | 0 | 200 | 1500 |
            | Ping Pong | 50,000 | 0 | 0 | **27.6** | 50 | 1000 |
            | **Hammer Nail** | 30,000 | 99.6 | 98 | 84 | 200 | **150** |
        - The above table highlights several areas of improvement for our harder tasks.
            1. **Sample-Efficiency of RL Algorithms**: We include the results of our **easiest** task, `Hammer Nail`, where all 3 of our baselines learned near-perfect control policies with just 150 episodes of experience during training. For our harder tasks, our strongest baseline **CrossQ** required **10x the amount of episodic experience** during training to learn good tool manipulation policies, while other baselines struggled significantly to match this performance.
            2. **Off-Policy RL for Short-Horizon Dynamic Tasks**: In `Ping Pong`, a dynamic task where a paddle must quickly hit a ball, we observed empirically that CrossQ and SAC struggled to conservatively hit the Ping Pong Ball. In Figure 7 of our paper, the Failure case for `Ping Pong` showcases at t=3 the ball being hit too aggressively. This was a frequent problem with CrossQ and SAC, while PPO did not have these issues. We believe for short-horizon dynamic tasks, there is more to be investigated about why Off-Policy RL methods struggle compared to On-Policy RL methods.
            3. **Learning Reward Functions**: In our `Scoop Particle` and `Scoop Cube` tasks, our human-engineered reward function still required 300,000 steps of training (1500 episodes of experience) to learn good tool manipulation policies. This is large in-comparison to tasks like `Hammer Nail`, which can be learned in just 30,000 steps of training (150 episodes of experience). We believe that learning a reward function from videos, or from structured data such as graphs, could provide a better reward signal for tasks like `Scoop Particle` and `Scoop Cube`, and enable progress in area (1) (i.e. better reward functions can improve the sample-efficiency of RL algorithms).
    - For complete details of each task category’s reward function design, success conditions, and environment resets, please see Section A.4 in our new Appendix.


2. **Updated Website with New Visualizations and Figures**
    - We have updated our project website, [https://mujoco-manipulus.github.io](https://mujoco-manipulus.github.io/), to include visualizations of our work. This includes an Overview of our Simulation Framework (Figure 2 in our paper), videos of successful policy rollouts for each of our 16 tasks, and a results section with Success Rate curves for each task (Figure 9 in our paper).

---

> ### Author Response · Authors · 2024-12-01
> **General Response to Reviewers (2/3)**
>
> 3. **Updated Baselines**
>     - Related to Update #1, “Updated Results w/ New Dense Reward Functions”, we have updated 3 components of our baselines
>     1. **Component #1: RL Algorithm Implementations**
>         - Previously, we used Stable-Baselines3 for our results. However, our updated results now use RL algorithms from [CleanRL](https://github.com/vwxyzjn/cleanrl), a popular RL algorithm library with high-quality single-file implementations with research-friendly features.
>     2. **Component #2: Added CrossQ algorithm as Baseline**
>         - We added [CrossQ](https://github.com/adityab/CrossQ), a highly sample-efficient model-free RL algorithm which consistently provided competitive results across all of our tasks (not including `Ping Pong`, which we discussed briefly in our paper in Section 4.2 and in our rebuttal response in Section 1).
>     3. **Component #3: Removed RGB Training Results**
>         - We provide results for all baselines using State Inputs, and we removed RGB Training Results from our work. We made this decision because Visual RL has 2 limitations in our work:
>             1. **MuJoCo CPU Training Time / Limitations of MuJoCo XLA (MJX)**: 100,000 steps of training with state inputs takes 10 minutes of wall-time (roughly), and 300,000 steps of training with state inputs takes 30 minutes of wall-time. Unlike GPU simulated benchmarks, such as [Isaac Lab](https://isaac-sim.github.io/IsaacLab/main/index.html) and [ManiSkill](https://maniskill.readthedocs.io/en/latest/), which are both based on the NVIDIA PhysX physics simulation backend, we do not have a GPU simulated benchmark. MuJoCo XLA (MJX), a JAX-based implementation of MuJoCo which simulates physics on the GPU, **currently does not support batch RGB rendering**, and this limitation is an out-of-scope contribution for our work which prevents us from training fast Visual RL policies in MuJoCo simulation. Despite this limitation, we are committed to implementing our tasks with MuJoCo XLA (MJX) using the [Brax](https://github.com/google/brax) framework in the future, once Batch RGB rendering is supported with the (currently) experimental “[Madrona MJX](https://github.com/shacklettbp/madrona_mjx)” batch renderer.
>             2. **Upper Bound of Performance**: Our benchmark focuses on whether we can learn tool manipulation policies with RL, and we use state information from MuJoCo as inputs to our policies since policies trained with state inputs can often represent the upper bound of performance for a given baseline. In the context of our work, the upper bound of performance means “**which policy can achieve the highest success rate for a task in the fewest # of total episodes of experience during training”.** Referring to our Table above, a policy learned for `Scoop Particle` with CrossQ+State Inputs in 1500 episodes of training experience is the upper bound of performance for that task.
>
> 4. **Design Choices for Action Space and Observation Space:**
>     - Several reviews raised concerns about our simple action spaces, and details of our observation space. We agree that these design considerations are important, and have updated our paper to include a detailed discussion of these items:
>         1. Action Space and Observation Space details are provided for each task category in Section 3 of our paper.
>         2. In summary, Tool Manipulation tasks have several constraints in real-world settings, and by constraining our Action Spaces, we simplify the RL problem while still learning the correct skills associated with the tool.
>         3. We provide qualitative videos at [mujoco-manipulus.github.io](http://mujoco-manipulus.github.io) for all 16 of our tool manipulation tasks to showcase the effectiveness of our learned policies. Each task exhibits the correct behavior, and benefits from learning a simpler optimization problem by using a constrained action space design.

---

> ### Author Response · Authors · 2024-12-01
> **General Response to Reviewers (3/3)**
>
> 5. **Differentiating our Simulation Benchmark from Existing Works**:
>     - Our benchmark is a first-of-its-kind tool manipulation benchmark. This is a critical area in robotics since it enables robots to make use of external items to accomplish tasks that are difficult with native hardware.
>     - In comparison to other benchmarks, such as Robosuite, Meta-World, and ManiSkill2, we provide our users with 16 tasks and 14 tools that are not included in these works, all in a unified setting.
>     - Please see Table 1 in our paper, which provides a comparison of our benchmark with existing simulation benchmarks, by comparing # of Tasks, Availability of Dense Rewards, # of Tool Skills in each benchmark, and Simulation Engine backend for each benchmark. We additionally provide a list of tool skills in each simulation benchmark, and compare against ours, in our Appendix in Section A.1.
>     - We hope this breakdown provides reviewers with a greater understanding of how our work sets a new bar for what can be achieved in simulation, and why our tool manipulation benchmark will enable greater research to be done in the areas of RL, Vision, and Robotics.
>
> 6. **Is Our Benchmark on the Pareto Frontier of Robotics?**
>     - **Speed of MuJoCo:**
>         - This point, raised by Reviewer eoGS, is important to address and we want to include this in our detailed general response to all reviewers. Unlike ManiSkill or Isaac Lab, which use NVIDIA PhysX to simulate physics, we use MuJoCo’s highly-optimized CPU backend. More specifically, we use MuJoCo 3.0+ in our benchmark, which we have plans to soon integrate with Brax and use the MuJoCo XLA (MJX) backend for GPU-simulated physics. At the time MuJoCo 3.0 was announced, the following results were published in [this discussion thread](https://github.com/google-deepmind/mujoco/discussions/1101) on Github:
>
>
>             | Platform | Time to train PPO at 60M Steps |
>             | --- | --- |
>             | MJX + Brax on TPU v5e-8 | 249 seconds |
>             | MJX + Brax on Nvidia A100 GPU | 718 seconds |
>             | Nvidia Isaac Gym on A100 GPU | 600 seconds |
>         - These results are from the following experiment “…Google’s [Barkour robot](https://blog.research.google/2023/05/barkour-benchmarking-animal-level.html) [learned] to run using MJX and [Brax](https://github.com/google/brax)’s PPO trainer. We train a policy that can be deployed on a real robot and compare the training time to the [Barkour publication](https://arxiv.org/abs/2305.14654), which used Nvidia Isaac Gym.” With the MJX + Brax Barkour environment, and using a TPU, PPO trained for 60 Million Steps in 249 seconds (4 minutes and 9 seconds) vs Isaac Gym taking 600 seconds on an A100 GPU (10 minutes).
>         - Since our environments are all implemented in MuJoCo, and will soon be integrated with MJX + Brax when Batch RGB rendering is supported for fast Visual RL, we will be the first MuJoCo-based and GPU-based simulation benchmark to support the variety of tasks in our task distribution, all with GPU accelerated physics, and we will be faster than Isaac Gym (based on NVIDA PhysX backend).
>     - **Sim-to-Real Potential and Accuracy of MuJoCo:**
>         - In a [recent work](https://sites.google.com/view/cloth-sim2real-benchmark) which benchmarked the Sim-to-Real capabilities of policies trained in **MuJoCo**, **Bullet**, **NVIDIA** **Flex**, and **SOFA**, the authors made the following conclusion from their extensive experiments:
>
>             > “Given the lower distances in both dynamic and quasi-static manipulation tasks shown by MuJoCo, as well as its capability of integrating robotic models, and availability of both CPU and GPU acceleration, **we recommend MuJoCo for learning cloth manipulation tasks in a simulation engine**.”
>             >
>         - In future updates of our work with tool manipulation, we plan to incorporate dynamic manipulation tasks, and will be using the current-best simulation benchmark for accurately simulating dynamic bodies and transferring the learned policies to real-world robots.
>
> We hope this summary of updates has addressed as many of the reviewers’ concerns as possible. For specific reviewer questions, we also provide reviewer-specific responses. We once again thank all the reviewers for their valuable feedback.

---

### Meta-Review · Area_Chair_aLCC · 2024-12-23

**Metareview:**

This paper proposed a benchmark designed for robotic tool manipulation using the MuJoCo physics engine. It was reviewed by four field experts and received unanimously negative evaluations. The main concerns raised include a lack of significant technical contributions and weak results. The AC finds no reasons to recommend acceptance. The authors are encouraged to address the reviewers' feedback and refine the work for submission to other venues.

**Additional Comments On Reviewer Discussion:**

The main concerns are most on limited technical novelty and weak experiments.

---

### Decision · Program_Chairs · 2025-01-22

Reject